



**Skill assessment of global, regional and coastal circulation forecast models:**
**evaluating the benefits of dynamical downscaling in IBI surface waters.**
**Pablo Lorente[1], Marcos García-Sotillo[1], Arancha Amo-Baladrón[1], Roland Aznar[1],**
**Bruno Levier[2], José Carlos Sánchez-Garrido[3], Simone Sammartino[3], Álvaro De**
**Pascual[1], Guillaume Reffray[2], Cristina Toledano[1] and Enrique Álvarez-Fanjul[1]**
[1]{Puertos del Estado, Madrid, Spain}
[2]{Mercator Ocean, Toulouse, France}
[3]{Physical Oceanography Group of University of Málaga (GOFIMA), Málaga, Spain}
Correspondence to: P. Lorente (plorente@puertos.es)
**Abstract**
In this work, a multi-parameter inter-comparison of diverse ocean forecast models was
conducted at the sea surface, ranging from global to local scales in a two-phase strategy.
Firstly, a comparison of CMEMS-GLOBAL and the nested CMEMS-IBI regional system was
performed against satellite-derived and in situ observations. Results highlighted the overall
benefits of both the GLOBAL data assimilation in open-waters and the increased horizontal
resolution of IBI in coastal areas, respectively. Besides, IBI proved to capture shelf dynamics
by better representing the horizontal extent and strength of a river freshwater plume,
according to the results derived from the validation against in situ observations from a buoy
moored in NW Spain. Secondly, a multi-model inter-comparison exercise for 2017 was
performed in the Strait of Gibraltar among GLOBAL, IBI and the nested SAMPA high-
resolution coastal forecast system in order to elucidate the accuracy of each system to
characterize the Atlantic Jet (AJ) inflow dynamic. A quantitative validation against High
Frequency radar (HFR) hourly currents highlighted both the steady improvement in AJ
representation in terms of speed and direction when zooming from global to coastal scales
though a multi-nesting model approach and also the relevance of a variety of factors at local
scale such as a refined horizontal resolution, a tailored bathymetry and a higher spatio-
temporal resolution of the atmospheric forcing. The ability of each model to reproduce a 2-
day quasi-permanent full reversal of the AJ surface inflow was examined in terms of wind-
induced circulation patterns. SAMPA appeared to better reproduce the reversal events
detected with HFR estimations, demonstrating the potential added value of coastal models
with respect to coarser parent regional systems. Finally, SAMPA coastal model outputs were
also qualitatively analysed in the Western Alboran Sea to put in a broader perspective the
context of the onset, development and end of such flow reversal episodes.
**Keywords:** forecasting; model; inter-comparison; validation, downscaling; HF radar; skill
assessment;





## 1   Introduction

Over the last three decades, significant progresses have been made in the discipline of operational oceanography thanks to the substantial increase in high-performance computational resources, which has fostered the seamless evolution in ocean modelling techniques and numerical efficiency (Cotelo et al., 2017) and given rise to an inventory of operational ocean forecasting systems (OOFSs) running in overlapping regions in order to reliably portray and predict the ocean state and its variability at diverse spatio-temporal scales.

Global circulation models have been steadily evolving in terms of complexity, horizontal resolution refinement and process parameterisation (Holt et al., 2017). Notwithstanding, such development involves compromises of scale and is subject to practical limits on the feasible spatial resolution (Greenberg et al., 2007). Although large-scale processes and physical and biogeochemical cycles are properly resolved by the current state-of-the-art global models resolution (e.g. nominal 1/12°), coastal and shelf phenomena are still poorly replicated or even misrepresented as the grid mesh is too coarse, especially for complex-geometry regions such as sea straits, archipelagos or semi-enclosed seas where the coastline, seamounts and bottom topography are not well resolved. In this context, tides, vertical coordinates, mixing schemes, river inflows and atmospheric forcings have been traditionally identified as five areas of further research in global ocean modelling (Holt et al., 2017).

Since the continental shelf is affected not only by natural agents (land-sea breezes, riverine discharges, bottom topography, coastline shape, etc.) but also by human-induced factors, an increased understanding of coastal circulation is essential for decision- and policy-making in the socioeconomically vital and often environmentally stressed coastal regions. Therefore, small-scale ocean features must be explicitly computed and accurately reproduced by means of regional models with finer horizontal grid spacing but for a particular delimited area. The success of this approach requires the seamless progress in several aspects, as previously identified by Kourafalou et al. (2015): i) a deep comprehension of the primary mechanisms driving coastal circulation; ii) downscaling methods to adequately represent air-sea and land-sea interactions; iii) robust methods to embed high-resolution models in coarser-scale systems. Therefore, this approach implies the transfer of large-scale information from the global model to the interior of the nested regional domain by means of diverse methodologies. One of them is the so-called 'spectral nudging' technique, adopted to ensure that the prevailing global conditions are not degraded in the open-ocean, while allowing sub-mesoscale processes to be resolved exclusively by the embedded model in the continental shelf and coastal areas (Herbert et al., 2014). Additionally, the regional modelling strategy can include some fine-tuning of physical parameters, individually tailored to each chosen area, instead of the universally valid parameterizations associated with global OOFSs.

The benefits of regional modelling over the driving global OOFS are generally assumed, but to date only few studies have explored and quantified the potential added value of such approach (Katavouta and Thomson, 2016; Rockel, 2015; Greenberg et al., 2007). The 'parent-son' model inter-comparison is mandatory during both implementation and operational stages since it aids to: i) verify the most adequate nesting strategy; ii) check the consistency of the nested model solution; and iii) identify any potential problem that might be inherited from the coarser system.

In the framework of the Copernicus Marine Environment Monitoring Service (CMEMS), a global ocean model together with a wealth of nested regional OOFSs are currently running in different areas of the European seas and providing paramount oceanographic forecast



products (Le Traon et al., 2018). Since the validation of OOFSs against independent
measurements constitutes a core activity in oceanographic operational centres, the skill of
Iberia-Biscay-Ireland (IBI) regional OOFS is routinely assessed by means of the NARVAL
(North Atlantic Regional VALidation) system (Sotillo et al. 2015), a web-based toolbox that
provides a series of skill metrics automatically computed and delivered in the QUality
Information Document - QUID - (Sotillo et al., 2014). In this context, the first goal of this
paper is to conduct a multi-parameter model inter-comparison between IBI regional OOFS
and the coarser parent system, the CMEMS GLOBAL (Lellouche et al., 2018), with the aim
of assessing their performance at the upper-layer. Their predictive skills to properly represent
the surface temperature (SST) over IBI coverage domain and diverse sub-regions were
evaluated by means of comparisons against remote-sensed and in situ observations. On the
other hand, their prognostic capabilities to accurately reproduce the coastal surface circulation
were assessed through the analysis of a single impulsive-type river outflow episode that took
place in March 2018 in the Galician coast (NW Spain), a region of freshwater influence -
ROFI- (Simpson, 1997).
Despite the recent advances in the development of CMEMS global and regional core
products, many downstream services for user uptake require information on even smaller
spatial scale, such as ocean forecasting for small island chains (Caldeira et al., 2016), intricate
bights (Stanev et al., 2016) or port approach areas where sharp topo-bathymetric gradients
pose special difficulties for accurate local predictions (Sotillo et al. 2018, under review;
Hlevca et al., 2018; Federico et al., 2017; Sánchez-Arcilla et al., 2016; Sammartino et al.,
2014; Grifoll et al., 2012). A variety of operational products for harbours have been recently
developed, although most of these coastal applications are wave and water-level forecasting
systems (Lin et al., 2008; Pérez et al., 2013). By contrast, lower attention has being devoted to
harbour hydrodynamic conditions since its reduced dimensions and intricate layout confer
upon harbour restrictions, which are not present in the open sea. Besides, derivative products
based on current forecasts, such as float trajectories, residence time maps, flushing patterns
and risk assessment of water quality degradation can constitute additional assets for efficient
harbour management (Álvarez-Fanjul et al., 2018; Sammartino et al., 2018). In order to
overcome the existing gap between the scales effectively solved by the regional OOFSs and
the coastal scales required to meet strong societal needs in support of blue and green growth,
a number of downstream services are currently adopting different downscaling approaches.
Dynamical downscaling takes regional boundary conditions to drive a high-resolution limited-
area model in which coastal processes are calculated on a finer grid by resolving well-known
hydrodynamic equations. However, uncertainties in the downscaling process are hard to
quantify since coastal solutions are still exchanging poorly controlled information with larger-
scale at their boundaries (Hernández et al., 2018).
As a representative example of downstream service developed by Puertos del Estado (PdE)
in a hot spot area like the Strait of Gibraltar (GIBST), the operational PdE-SAMPA high-
resolution coastal system (Sánchez-Garrido et al., 2013) is dynamically embedded in IBI and
nowadays employed by the Port Authority of Algeciras Bay as predictive tool to support
maritime policy and assist high-stakes decision-making related to marine safety, port
operation optimization and mitigation of both natural disasters and anthropogenic hazards.
Previous researches have unequivocally proved the ability of PdE-SAMPA to accurately
capture basic circulation features of the GIBST area and Algeciras Bay (Sanchez-Garrido et
al., 2014; Sammartino et al., 2014; Soto-Navarro et al., 2016). A preliminary model skill
assessment was conducted within the framework of MEDESS-4MS project (Sotillo et al.,
2016-a). However, the added value of this coastal OOFS with respect to its parent regional





system IBI was only quantified from a lagrangian perspective by using a wealth of drifters.
The second goal of this contribution is thus to build up upon previous model inter-comparison
exercises, placing special emphasis on the characterization of the Atlantic Jet (AJ) inflow into
the Mediterranean Sea in terms of speed and direction. This geostrophically adjusted jet
fluctuates in a wide range of temporal scales and drives the main circulation in the Alboran
Sea, feeding and surrounding the Western Alboran Gyre -WAG- (Macias et al., 2016). An
inter-comparison exercise was conducted for 2017 among a global configuration (CMEMS
GLOBAL), a regional application (CMEMS IBI) and a higher resolution coastal system (PdE-
SAMPA), in order to characterize the AJ dynamics and their ability to adequately capture an
extreme event: the quasi-permanent (up to ~48 h long) full reversal of the AJ surface flow
under intense and prolonged easterlies. To this end, a High-Frequency radar (HFR) has been
used as benchmark since it regularly provides quality-controlled hourly maps of the surface
currents of the Strait (Lorente et al., 2014).
In summary, this paper serves one primary purpose: performing a multi-parameter model
skill assessment in IBI surface waters, ranging from global to local scales in a two-phase
strategy: i) a comparison between GLOBAL and IBI regional systems in the entire
overlapping coverage domain, posing special attention on regionalization; and ii) a process-
based multi-model inter-comparison for 2017 with a focus on the GIBST. It is worth
mentioning that the present model inter-comparison is limited to the surface layer.
This paper is organized as follows: Section 2 provides further details about the study areas.
Section 3 describes the diverse models configuration. Section 4 outlines the observational
data sources and methodology used in this study. Sections 5 and 6 present a detailed
discussion of the results. Finally, main conclusions are summarized in Section 7.
**2    Study Areas**
**2.1    IBI area (and subregions)**
From a pure physical oceanographic point of view, the IBI geographical domain is a very
complex region (Figure 1, a), marked by a generally steep slope separating the deep ocean
from the shelf. The western, and deeper, side of the IBI domain is affected by main large-
scale currents, mainly the closure of the North Atlantic Drift, here split into two major
branches, the major one continuing northwards along the north-western European shelves
(NAC and NADC) and the other, the Azores Current (AC), which follows south-eastwards
and has continuity in the Canary Current (CaC). On the other hand, along the slope, a
poleward slope current flows in the subsurface; it is observed as far north as at Ireland
latitudes. Instabilities in this slope current favour the occurrence of slope water oceanic
eddies, along the northern Iberian coast (Pingree & Le Cann 1992). On the continental
shelves, intense tidal motions provide the dominant source of energy (Álvarez-Fanjul et al.
1997): noticeable tidal mixing fronts arise on the most energetic tidal areas of the IBI region
(i.e. English Channel, Celtic and Irish Sea). Shelf and coastal areas of the region are also
affected by strong storm surges (Pérez et al. 2012). Along the western Iberian and African
coasts, strong summer upwelling of bottom cold and enriched waters take place under
predominant northerly wind conditions that trigger the Ekman-driven offshore deflection of
the surface flux.
IBI is also a rather broad and heterogeneous area. In order to gain insight into the model
skill assessment (as later exposed in Section 5), IBI service (IBISR) regional domain has been
split in nine different subregions (Figure 1-a): the Irish Sea (IRISH), the English Channel



(ECHAN), the Gulf of Biscay (GOBIS), the North Iberian Shelf (NIBSH), the West Iberian
Shelf (WIBSH), the Western Mediterranean Sea (WSMED), the Gulf of Cadiz (CADIZ), the
Strait of Gibraltar (GIBST) and the Canarias Islands (ICANA).

## 2.2   Strait of Gibraltar

The Strait of Gibraltar (GIBST), the only connection between the semi-enclosed
Mediterranean basin and the open Atlantic Ocean, is characterized by a two-layer baroclinic
exchange which is hydraulically controlled at Camarinal Sill (Figure 1, b). Whilst saltier
Mediterranean water flows out at depth, an eastward surface jet of relatively fresh Atlantic
water (AJ) flows into the Alboran Sea by surrounding the quasi-permanent Western
Anticyclonic Gyre (WAG) and the more elusive Eastern Anticyclonic Gyre (EAG) in a
wavelike path. As the WAG owes its existence to the input of new Atlantic waters provided
by the AJ, both structures are widely considered to be coupled and usually referred as to the
AJ-WAG system. A significant variety of analytical, field and modelling studies have
previously attempted to disentangle the AJ-WAG system and properly explain the underlying
physical processes (Sánchez-Garrido et al., 2013; Macías et al., 2007-a; Viúdez, 1997).
The position, intensity and direction of the AJ fluctuate in a broad range of temporal
scales, driving the upper-layer circulation of the Alboran Sea with subsequent physical and
biological implications (Solé et al, 2016; Sánchez-Garrido et al. 2015; Ruiz et al., 2013). For
instance, the presence of a strong AJ close to the northern shore of the Alboran Sea reinforces
the coastal upwelling and therefore increases both the near-shore chlorophyll concentration
and the spawning of fish in this region (Ruiz et al., 2013; Macías et al., 2008). By contrast,
meteorologically-induced inflow interruptions can trigger the weakening and even the
decoupling of the AJ-WAG system (Sánchez-Garrido et al., 2013), the subsequent eastward
migration of the WAG and the genesis of a new gyre that coexists with the other two, giving
rise to a three-anticyclonic-gyre situation (Viúdez et al., 1998).
Within this context, the AJ pattern has been described to oscillate between two main
circulation modes at seasonal scale (Vargas-Yáñez et al., 2002): i) a stronger AJ flows north-
eastwards during the first half of the year and ii) a weaker AJ flows more southwardly
towards the end of the year. Sea level Pressure (SLP) variations over the Western
Mediterranean basin and local zonal wind (U) fluctuations in the Alboran Sea have been
usually considered as the main factors controlling and modulating the AJ variability (Macías
et al., 2007-b; Lafuente et al., 2002). In particular, the second parameter has been largely
invoked as the primary driving agent to explain both the intensification of the surface inflow
during prevalent westerlies and also extreme AJ collapse events recorded when intense
easterlies are predominant (Macías et al., 2016). The zonal wind intensity has been reported to
follow an annual cycle with more westerly (easterly) winds during winter (summer) months
(Dorman et al., 1995). The seasonal variability and occasional interruptions of the Atlantic
inflow due to meteorological forcing have been earlier investigated with in situ data from
fixed moorings (García-Lafuente, 2002). More recently, a considerable number of satellite
tracked drifters were released on both sides of GIBST within the framework of MEDESS-
4MS project, providing hence a complete Lagrangian view of the Atlantic waters inflow into
the Alboran Sea (Sotillo et al., 2016-b).



## 3    Models description

Whereas basic features of the three OOFSs employed in this work are gathered in Table 1, further details are provided in the following devoted sub-sections.

### 3.1    CMEMS GLOBAL system

The Operational Mercator global ocean analysis and forecast system provides 10 days of 3D global ocean forecasts updated daily. This product includes daily mean files of temperature, salinity, currents, sea level, mixed layer depth and ice parameters from the surface to seafloor over the global ocean. It also includes hourly mean surface fields for sea level height, temperature and currents. The global ocean output files are displayed with a 1/12 degree horizontal resolution with regular longitude/latitude equirectangular projection. 50 vertical levels span from 0 to 5500 meters.

The system is based on the Nucleus for European Modelling of the Ocean (NEMO) v3.1 ocean model (Madec, 2008). The physical configuration is based on the tripolar ORCA grid type with a horizontal resolution of 9 km at the equator, 7 km at Cape Hatteras (mid-latitudes) and 2 km toward the Ross and Weddell seas. The 50-level vertical discretization retained for this system has 1 m resolution at the surface decreasing to 450 m at the bottom, and 22 levels within the upper 100 m. The bathymetry used in the system is a combination of interpolated ETOPO1 and GEBCO8 databases. The system was initialized on 11 October 2006 based on the temperature and salinity profiles from the EN4 monthly gridded climatology. The atmospheric fields forcing the ocean model are taken from the ECMWF (European Centre for Medium-Range Weather Forecasts) Integrated Forecast System. A 3-h sampling is used to reproduce the diurnal cycle. The system does not include neither tides nor pressure forcing. The monthly runoff climatology is built with data on coastal runoffs and 100 major rivers from the Dai et al (2009) database (Lellouche et al., 2018). Altimeter data, in situ temperature and salinity vertical profiles and satellite sea surface temperature are jointly assimilated to estimate the initial conditions for numerical ocean forecasting. Moreover, satellite sea ice concentration is now assimilated in the system in a monovariate/monodata mode. Further details can be found in Lellouche et al., (2018).

### 3.2    CMEMS IBI regional system

The IBI OOFS provides a real-time short-term 5-day hydrodynamic 3D forecast (and one day of hindcast as best estimate) of a range of physical parameters (currents, temperature, salinity and sea level) since 2011 (Sotillo et al., 2015). IBI is based on an eddy-resolving NEMO model application (v3.6) that includes high-frequency processes required to characterize regional-scale marine processes. The model application is run at 1/36° horizontal resolution and final products are routinely delivered in a service domain extending between 19°W-5°E and 26°N-56°N. The NEMO model (Madec, 2008) solves the three-dimensional finite-difference primitive equations in spherical coordinates discretized on an Arakawa-C grid and 50 geopotential vertical levels (z coordinate), assuming hydrostatic equilibrium and Boussinesq approximation. Partial bottom cell representation of the bathymetry (a composite of ETOPO 2 and GEBCO8) allows an accurate representation of the steep slopes characteristic of the area. The model grid is a subset of the Global 1/12° ORCA tripolar grid used by the parent system (the CMEMS GLOBAL system) that provides initial and lateral boundary conditions, but refined at 1/36° horizontal resolution.

The IBI run is forced every 3 hours with up-to-date high-frequency (1/8° horizontal grid resolution) meteorological forecasts (10-m wind, surface pressure, 2-m temperature, relative humidity, precipitations, shortwave and longwave radiative fluxes) provided by ECMWF.





CORE empirical bulk formulae (Large and Yeager, 2004) are used to compute latent sensible heat fluxes, evaporation and surface stress. Lateral open boundary data are interpolated from the daily outputs of the GLOBAL system. These are complemented by 11 tidal harmonics built from FES2004 (Lyard et al., 2006) and TPXO7.1 (Egbert and Erofeeva, 2002) tidal models solutions. Fresh water river discharge inputs are implemented as lateral open boundary condition for 33 rivers. Flow rate data imposed is based on a combination of daily observations from PREVIMER, simulated data from E-HYPE hydrological model and monthly climatological data from GRDC and French "Banque Hydro" dataset. Further details can be found in Sotillo et al., (2015).

Originally, the operational IBI system was based on a periodic re-initialization from the GLOBAL parent solution. Afterwards, IBI has steadily evolved: by April 2016, an upgrade of the downscaling methodology was implemented, substituting the periodic re-initialization by an spectral nudging technique in order to avoid temporal discontinuity inherent to the periodic re-initialization and minimize dependency from the GLOBAL parent solution on the shelf. The chosen spectral nudging method permits to ″nudge″ the low frequency IBI system solution towards the large scale GLOBAL solution in those areas where this global solution is supposed to be better (mainly off the shelf and in deep waters) due to the assimilation of lower frequency signals. Thus, the nudging is regionally limited in those areas where the parent system can not improve the regional model (e.g. where there is no data assimilation of altimetry or where the physics is missing, for instance on the shelf) or where the spatial filtering processes are potentially damage (for instance close to the bottom or the open boundaries). This spatial weight function is a 3D mask showing transitions between zones where the IBI system is kept nudged (typically off shore, outside from the shelf area, and off the open boundaries) and the ones where IBI remains free. Finally, a SAM2-based data assimilation scheme (Lellouche et al., 2013; Brasseur et al., 2005) was recently introduced (April 2018) in order to enhance IBI predictive skills but will not be further described here as only outputs from 2017 have been used in the present work.

### 3.3 PdE SAMPA coastal system

The PdE-SAMPA operational forecast service started in April 2012 (Sammartino et al., 2014; Sánchez-Garrido et al., 2014). It routinely provides a daily short-term forecast (72-h horizon) of currents and other oceanographic variables in the Gibraltar Strait and its surroundings (Gulf of Cadiz and Alboran Sea). The PdE-SAMPA model application was developed by the University of Malaga in collaboration with PdE and it is based on the Massachusetts Institute of Technology global circulation model -MITgcm- (Marshall et al., 1997). The domain, which extends from the Gulf of Cádiz to the Alboran Sea, is discretized with an orthonormal curvilinear grid of variable horizontal resolution, sparser close to the boundaries (~ 8-10 km) and higher in GIBST (~ 300-500 m). In the vertical dimension, SAMPA has 46 unevenly spaced z levels with maximum resolution of 5 m near the surface, exponentially decaying towards the seafloor. The shallower level is at 2.5 m depth. The bathymetry is derived from a combination of the GEBCO bathymetry data set and fine-resolution bathymetric charts of the Strait of Gibraltar and the continental shelf of the Gulf of Cadiz and northern coast of the Alboran Sea. The bottom topography is represented as partial vertical cells. In the two lateral open boundaries (west and east) the model is forced by daily mean temperature, salinity and velocity fields from CMEMS-IBI regional model (Sotillo et al., 2015). In addition, tidal and meteorologically-driven barotropic velocities are prescribed across the open boundaries, the former extracted from the model described by Carrere and Lyard (2003) and the latter from the storm surge operational system developed by Álvarez-Fanjul et al. (2001), which accounts for the remote effect of the atmospheric forcing in the



barotropic flow through GIBST. At the sea surface, the model is forced by hourly values of
wind stress, air humidity and temperature, fresh water and heat surface fluxes provided by the
Spanish Meteorological Agency through the operational Forecast System based on the
HIRLAM model (Cats, G.; Wolters, 1996). Further details on the SAMPA model
configuration are provided in Sanchez-Garrido et al. (2013).

## 4 Validation of OOFSs

### 4.1 Framework

The validation of OOFSs against independent measurements constitutes a core activity in
oceanographic operational centres since it aids: i) to infer the relative strengths and
weaknesses in the modelling of several key physical processes; ii) to compare different
versions of the same OOFS and evaluate potential improvements and degradations before a
new version is transitioned into operational status; iii) to compare coarse resolution 'father'
and nested high-resolution 'son' systems to quantify the added value of downscaling.
With regards to the third aspect, IBI forecast products are regularly intercompared not only
against other CMEMS regional model solutions (e.g. NWS and MED) in the overlapping
areas (Lorente et al., 2017) but also against its parent system (GLOBAL) by means of
NARVAL (North Atlantic Regional VALidation) login-protected web-based application
(Sotillo et al. 2015). This tool has been implemented to routinely monitor IBI performance
and to objectively inter-compare models' reliability and prognostic capabilities. Both real-
time validation ('online mode') and regular-scheduled 'delayed-mode' validation (for longer
time periods) are performed using a wealth of observational sources as benchmark, among
others: in situ observations from buoys and tide-gauges, SST satellite derived products,
temperature and salinity profiles from ARGO floats and HFR. Product quality indicators and
skill metrics are automatically computed in order to infer IBI accuracy and the spatiotemporal
uncertainty levels. The evaluation metrics regularly generated by NARVAL are online
delivered in the QUID, which is periodically updated and freely available in CMEMS website
(http://marine.copernicus.eu/).
Complementarily, opportunistic inter-comparisons are conducted in the frame of diverse
EU-funded projects such as MEDESS-4MS (Sotillo et al., 2016-a): 35 satellite tracked drifters
were released on both sides of the Strait of Gibraltar and the quality-controlled in situ data of
sea surface temperature and currents were collected to build the MEDESS-GIB database
(Sotillo et al., 2016-b), providing hence a complete Lagrangian view of the surface inflow of
Atlantic waters through the GIBST and the Alboran Sea. Such valuable oceanographic
information was subsequently used to intercompare IBI and SAMPA forecast products to
identify strengths (realistic simulation of the Atlantic Jet and the Algerian Current) and
shortcomings (position and intensity of the Alboran gyres, especially the western one) in both
models performance. This exercise reflected the effectiveness of the dynamical downscaling
performed through the SAMPA system with respect to the regional solution (in which
SAMPA is nested), providing an objective measure of the potential added value introduced by
SAMPA.
Eventually, ancillary validation approaches have been recently adopted focused on the
evaluation of ocean models performance in specific situations and on their ability to
accurately reproduce singular oceanographic processes (Hernández et al., 2018). Since the
NARVAL tool is devoted to inter-compare model solutions on a monthly, seasonal or annual
basis, part of the picture is missing due to traditional time averaging. Hence the quality


indicators computed, albeit valid, mask somehow models´ capabilities to replicate ocean
phenomena of particular interest at shorter timescales. This event-oriented multi-model inter-
comparison methodology allows to better infer the ability of each system to capture small-
scale coastal processes. In this context, the recurrent question *"Which model is the best one?"*
should be reformulated by firstly admitting that one system can outperform the rest of OOFs
for a particular event but by contrast can be also beaten when attempting to reproduce and
characterize some other distinct ocean phenomenon.
Those oceanographic events subject of further insight might encompass, among others: i)
coastal upwelling, dowelling and relaxation episodes; ii) submesoscales eddies (Mourre et al.,
2018); iii) extreme events; iv) complete flow reversals. Particularly, in the present work the
attention has been devoted to the full and permanent reversal of the surface AJ in the GIBST
during, at least, 48 hours. This unusual episode has been detected by means of HFR current
estimations and further examined with OOFSs outcomes. The agreement between both in situ
and remote-sensing instruments and the ocean forecasting system has been evaluated by
means of computation of a set of statistical metrics traditionally employed in this framework:
histograms, bias, root mean squared differences (RMSD), scalar and complex correlation
coefficients, current roses, histograms, quantile-quantile (QQ) plots and the best linear fit of
scatterplots. In the following sub-section all the in situ and remote-sensed observations
employed in the present work are described.
**4.2  Observational data sources**
*In situ observations*
The study domain includes an array of buoys operated by Puertos del Estado and the Irish
Marine Institute (Figure 1, a), providing quality-controlled hourly-averaged observations of
SST, SSS and currents. To ensure the continuity of the data record, occasional gaps detected
in time series (not larger than 6 hours) were linearly interpolated. Basic features of each in-
situ instrument are described in Table 2.
*Satellite-derived observations*
The European Ocean Sea Surface Temperature L3 Observations is a CMEMS operational
product which provides a daily fusion of SST measurements from multiple satellite sensors
over a 0.02° resolution grid. The L3 multi-sensor (supercollated) product is built from bias-
corrected L3 mono-sensor (collated) products. If the native collated resolution is N and N <
0.02 the change (degradation) of resolution is done by averaging the best quality data. If N >
0.02 the collated data are associated to the nearest neighbour without interpolation nor
artificial increase of the resolution. A synthesis of the bias-corrected L3 mono-sensor
(collated) files remapped at resolution R is done through a selection of data based on the
following hierarchy: AVHRR_METOP_B, SEVIRI, VIIRS_NPP, AVHRRL-19, AVHRRL-
18, MODIS_A, MODIS_T, AMSR2. This hierarchy can be changed in time depending on the
health of each sensor. Further details can be found in the Product User Manual (PUM), freely
available in CMEMS website (http://cmems-resources.cls.fr/documents/PUM/CMEMS-SST-
PUM-010-009.pdf)
*HFR-derived observations*
The HFR system employed in the present study consists of three-site shore-based
CODAR Seasonde network, installed in GIBST (Fig 1, b-c). Hereafter the sites will be
referred to by their four letter site codes: CEUT, CARN, and TARI, respectively (Figure 1, c).
Each site is operating at a central frequency of 26.8 MHz, providing hourly radial current
measurements which are representative of the upper 0.5 m of the water column. The



maximum horizontal range and angular resolution are 40 km and 5º, respectively. Radial
current measurements from the three stations are geometrically combined with an averaging
radius set to 3 km, in order to estimate hourly total current vectors on a Cartesian regular grid
of 1x1 km horizontal resolution.

5       A source of error to be considered in the computation of the total vectors is the so-called
Geometrical Dilution of Precision (GDOP). The GDOP is defined as a dimensionless
coefficient of uncertainty that characterizes how radar system geometry may impact on the
measurements accuracy and position determination errors, owing to the angle at which radial
vectors intersect. Maps of east and north GDOP for this HFR system (not shown) follow a
pattern where their values increase with the distance from the radar sites and along the
baselines (lines connecting two HFR sites), as the combining radial vectors are increasingly
parallel and the orthogonal component tends to zero. Further details can be obtained from
Lorente et al. (2018).
The accuracy of HFR measurements, which are affected by intrinsic uncertainties (radio
frequency interferences, environmental noise, etc.) have been previously assessed by
comparing against in situ observations provided by a point-wise current meter (Lorente et al.,
2014), yielding correlations above 0.7 and RMSD below 13 cm·s$^{-1}$. Such results revealed that
this HFR network has been operating within tolerance ranges, properly monitoring the surface
circulation in near real-time of this geostrategic region.
Recent works relying on this HFR system have successfully investigated the water
exchange between Algeciras Bay and the Strait of Gibraltar (Chioua et al., 2017), the impact
of the atmospheric pressure fluctuations on the mesoscale water dynamics of the Strait of
Gibraltar and the Alboran Sea (Dastis et al., 2018), the dominant modes of spatio-temporal
variability of the surface circulation (Soto-Navarro et al., 2016) or the characterization of the
Atlantic surface inflow into the Mediterranean Sea (Lorente et al., 2018).
In the present work, quality-controlled hourly HFR current measurements collected during
the entire 2017 were used as benchmark to elucidate the skill of a number of OOFSs. As
shown in Figure 1-c, the data availability was significantly high: almost 100% in the selected
transect, decreasing in the easternmost sectors. The transect here used to examine the AJ
surface inflow was readily chosen as the associated total GDOP, reported in Lorente et al
(2018), was reduced (below 1.3) and the spatial and temporal data availability were optimal
during 2017. From an oceanographic perspective, the election of such transect was also
convenient to better characterize both the intensity and direction of the AJ, since its midpoint
covers the area where the highest peak of current speed is usually detected and also where the
inflow orientation is not influenced yet by the water exchange between Algeciras Bay and the
Strait of Gibraltar.
**5    Comparison between CMEMS model solutions in IBI waters**
*Temperature*
The CMEMS L3 satellite-derived daily data were used to validate the SST fields predicted
by both GLOBAL and IBI. Maps of SST anomalies were computed on a monthly basis
(Figure 2). Apparently, both models behaved similarly during winter (defined as JFM). In
January, low warm anomalies were detected off-shelf in northwest Atlantic waters, while cold
SST anomalies were encountered in coastal areas of the English Channel and the Irish Sea,
and also surrounding the Balearic Islands in the Western Mediterranean (Figure 2, a-b). By
March, the moderately positive western anomalies migrated southwards within IBI domain



(Figure 2, c-d). The negative bias previously observed almost disappeared around the Balearic Islands and in the Irish Sea, turning even into slightly warm anomalies inside the English Channel in the case of GLOBAL.

During spring months (AMJ), a small positive bias spread over almost the entire IBI domain (Figure 2, e-f). In May, GLOBAL outputs exhibited a moderate SST overestimation (around 1ºC) in the English Channel, the Irish Sea and the southern part of the North Sea (Figure 2, e). In addition, a narrow belt of warm SST anomalies could be observed along the African coastal upwelling system (ACUS hereinafter) and over the continental shelf break (evident until late July). An increased vertical mixing during summer has been previously postulated to reduce the SST observed over the continental shelf break with respect to the surrounding ocean (Graham et al., 2018), thus explaining such strip of warm bias. By contrast, since IBI presents a higher grid resolution, it could partially resolve the internal waves breaking which leads to turbulence and energy for increased vertical mixing with cooler waters beneath the pycnoclyne, ultimately contributing to the reduced SST bias observed in IBI estimations over the continental shelf break (Figure 2, f). Besides, IBI properly reproduced the SST field in the northern IBI area although clearly overpredicted coastal temperatures over ACUS, in the periphery of the Canarias Islands.

During summertime (JAS), the warm SST spring anomalies previously identified in both GLOBAL and IBI became even more pronounced in July, locally reaching values up to 2ºC (Figure 2, g-h). Besides, new features were revealed as a result of the onset of the upwelling season, such as the positive bias in the Iberian Upwelling System (IUS), higher in the case of IBI (Figure 2, h). According to the reduced bias in both western coastal upwelling systems (Figure 2, g), GLOBAL seemed to better replicate the SST field likely thanks to recent progresses in data assimilation schemes and to the growing wealth of available observational data. In this context, future data sets (satellite SSS and swath SST) should improve constrains on model behaviour (Gasparin et al., 2018). By September, a portion of the warm anomalies detected in GLOBAL outputs vanished in the Irish Sea and the English Channel although a dipole of positive-negative anomalies could be clearly observed in the North Sea (Figure 2, i). By contrast, in the case of IBI the SST overestimation expanded over the entire ICANA sub-region (as defined in Figure 1, a) and the Gulf of Lion (Figure 2, j). Furthermore, the summer positive anomaly developed in the Western Alboran Sea became warmer, likely linked to the inadequate representation of the speed and direction of the Atlantic Jet, something that will be addressed in the following sections.

By the end of the year (Figure 2, k-l), both models appeared to better fit to observations as reflected by the smoothed SST anomalies in northern (southern) sub-regions observed for GLOBAL (IBI). GLOBAL underestimated again the SST over the Irish Sea and the English Channel (Figure 2, k) as it previously did in January (Figure 2, a), thus closing the cold-warm-cold anomalies cycle during 2017. The alternation between winter cold anomalies and summer warm anomalies were earlier identified by Graham et al. (2018) and related to a possible over-stratification in coastal regions. With regards to IBI predictions, a relevant cold bias was detected in the Western Mediterranean, particularly over the Algerian Current and the Almeria-Oran Front (Figure 2, l).

It is worth mentioning that availability of satellite observations is lower close to the shorelines and the associated intrinsic uncertainties are higher, probably due to the impact of the land mask on the optimal interpolation process conducted to transform the original satellite tracks into a regular grid, justifying to some extent the predominance of SST anomalies in coastal areas.



Complementary skill metrics, spatially-averaged over the entire IBI service (IBISR)
domain, were depicted on a monthly basis (Figure 3, a). While significantly high correlation
coefficients (above 0.95) remained rather constant for both OOFS, monthly RMSD exhibited
a marked seasonal cycle with lower values in winter (0.4ºC), a spring rise reaching a peak of
0.7ºC by July due to the aforementioned model SST overestimation, followed by a steady
decay during the last part of the year. GLOBAL performance was more accurate from July to
December as indicated by the lower RMSD.
According to the sub-regional monthly statistics (Figure 3, b-j), a rather similar sequence
could be also found for open-water zones such us the Gulf of Biscay (GOBIS, Figure 3-b) or
the Western Mediterranean (WSMED, Figure 3-c): permanently consistent correlations
(above 0.8) and higher RMSD for the second part of the year. In other sub-regions, the RMSD
evolved in like fashion while summer correlation values decreased down to [0.4-0.6] ºC: Gulf
of Cadiz (CADIZ, Figure 3-d), Canarias Islands (ICANA, Figure 3-i) and Strait of Gibraltar
(GIBST, Figure 3, j). In near-coast areas, skill metrics fluctuated differently: in the Western
Iberian Shelf (WIBSH, Figure 3-e), higher RMSD were observed in winter and summer as a
consequence of a likely misrepresentation of the freshwater discharge during the rainy season
and of the coastal upwelling under northerly wind regime, respectively. By contrast, in the
Northern Iberian Shelf (NIBSH, Figure 3-f) the monthly correlation coefficient oscillated
without a clear pattern, ranging from 0.4 (March and December) to 0.8 (early summer).
Furthermore, IBI seemed to outperform GLOBAL in the two northernmost sectors within the
study-domain: English Channel (ECHAN, Figure 3-g) and Irish Sea (IRISH, Figure 3-h) in
terms of higher correlation and lower RMSD, especially pronounced during summertime.
Tidally driven mixing could account for a portion of the discrepancies encountered between
the coarser detided GLOBAL and IBI model solutions, where the former predicts an over-
stratification in shelf-seas. Finally, as SST estimations are routinely assimilated in GLOBAL
system, a more precise performance is generally expected offshore, as proved for ICANA or
WESMED, among other sub-regions.
Results above exposed reveal that SST divergences between IBI and GLOBAL forecast
datasets are mainly found on the shelf near coastal areas featuring a complex bathymetry. For
the sake of completeness, supplementary validation works in the entire 3D water column with
Argo-floats are regularly conducted to assess model vertical structure (not shown, we refer the
reader to the QUID). Both models perform fairly well in open-waters, given the fact that
GLOBAL assimilates this type of in situ observations and subsequently transfers the
information to the nested IBI system thanks to the aforementioned spectral nudging
technique. Nevertheless, validation focused on smaller scales and high frequency processes is
still crucial to analyse in detail the performance of both modelled products in intricate coastal
regions.
Hourly in situ observations from eight buoys, moored within specific sub-regions (Figure
1, a), were used as benchmark to validate both GLOBAL and IBI outputs. The annual time
series of SST exhibited a significantly high resemblance, properly reproducing the expected
annual-cycle shape (Figure 4). According to the consistent skill metrics derived from the
comparison against three deep-water buoys (B3, B4 and B5, in Table 2), both models had a
rather alike performance during 2017 with RMSD and correlation coefficients in the ranges
[0.44-0.96] ºC and [0.86-0.99], respectively (Figure 4: c, d, e). While the similar behaviour
observed off the shelf is partially attributable to the aforementioned spectral nudging
technique, the model-observation comparison in near-shore areas revealed noticeable
discrepancies.



On one hand, IBI appeared to outperform GLOBAL system in the Irish Sea (Figure 4, b),
Gulf of Cadiz (Figure 4, f) and GISBT sub-region (Figure 4, g), as reflected by lower (higher)
RMSD (correlation) values obtained. Particularly, the results for the Strait of Gibraltar are not
in complete accordance with the statistics previously derived from the comparison against L3
satellite-derived data (Figure 3, j), likely due to the fact that remote-sensed SST estimations
area might be affected by higher intrinsic uncertainties (i.e. land contamination and cloud
cover). Although both comparisons against remote and in situ observations confirmed the
model SST overestimation in GIBST, especially during summertime, the former (latter)
indicated that IBI precision was significantly lower (higher). Another relevant aspect is the
notable ability of IBI to capture sharp summer SST drops (steeper than 3ºC) during prevalent
easterlies (Figure 4, g), as a result of the surface inflow reversal and subsequent intrusion of
warmer Mediterranean waters into GIBST (this phenomenon will be subject of further
analysis in Section 7). However, GLOBAL appeared to overestimate SST in this area during
the entire year, as reflected by a RMSD of 1.64ºC.
On the other hand, GLOBAL seemed to behave slightly better at B1 location -IBISR area-
(Figure 4, a) and substantially more accurately at B8 buoy location - in the Canarias Islands,
ICANA-, where a permanent SST overestimation from June to December was evidenced in
IBI predictions (Figure 4, h), yielding thereby a RMSD twice higher than that obtained for
GLOBAL estimations, in agreement with Figure 2 (i-j) and Figure 3 (i). The lower
performance of IBI in ICANA sub-region was previously reported by Aznar et al (2016) when
inter-comparing IBI forecast and 1/12º reanalysed solutions. At this point it is worth recalling
that GLOBAL includes a data assimilation scheme, whereas IBI takes realistic ocean
conditions from weekly global analyses. This fact shows up the possible benefits of the
observational data assimilation in these areas, at least in terms of surface variables.
Furthermore, a fraction of observed model-buoy discrepancies in SST can be explained in
terms of disparate depth scales: whereas IBI and GLOBAL daily outputs are representative of
the temperature in the upper one meter of the water column, moored buoys provide
temperature estimations at a deeper nominal depth (between 1 and 3.5 m, depending on the
brand).
Complementarily, a quarterly analysis was performed to infer any potential degradation in
model performances during a specific season of the year (Figure 5). Overall, both GLOBAL
and IBI predictions seemed to be more reliable in winter (except at B1 location: Figure 5-a) in
terms of lower RMSD. They also emerged to be less realistic during summer, as denoted by
abrupt decreases in quarterly correlation indexes (from 0.9 down to 0.5) at B2, B4 and B6
locations and the relevant rise of RMSD (up to 2.5ºC) at B7 location (GIBST sub-region).
This SST overestimation could be partially explained in terms of imprecise latent sensible
heat fluxes and excess of evaporation, although additional efforts should be devoted to shed
light on it. Once again, IBI performance appeared to be more accurate in coastal zones
featuring a more complex bathymetry (at B2, B4, B6 and B7 locations), whereas GLOBAL
fitted better to in situ observations in off shelf regions such as at B1 and B8 locations. In the
rest of the cases, both model solutions were rather alike. It is noteworthy that each point-wise
buoy is not representative of the entire sub-region in which is deployed, explaining thus to
some extent the discrepancies arisen between sub-sections 6.1 and 6.2.
*Salinity*
As pointed out in the introduction, the enhancement of riverine forcing is still as a priority
in ocean modelling as the estuarine circulation is mainly driven by horizontal density
gradients which are ultimately modulated by freshwater inputs. In this context, previous





works have investigated the potential benefits of replacing old climatologies by data from
hydrological model predictions (O´Dea et al., 2017). Here we provide a specific example to
illustrate the discrepancies between GLOBAL and IBI performances in the Galician coast
(NW Spain), as a consequence of the different horizontal resolution and distinct runoff
forcing implemented in the operational chain. While both models performances are rather
similar in open-waters (according to the results derived from the validation against 3D Argo-
float profiles and exposed in the QUID), higher discrepancies are expected to arise in coastal
and shelf areas as they are governed by small-scale processes such as land-sea breezes, runoff
(and the resulting stratification and buoyancy-driven circulation), transport materials
(nutrients, sediments, pollutants, etc.).
As shown in Figure 6-a, hourly in situ SSS data collected by B4 buoy during March 2018
experienced an abrupt decrease from a standard value around 36 PSU down to almost 33 PSU
in just few hours during the 20[th] of March, likely due to a noticeable filament of freshwater
discharged by Miño River. IBI outputs at the closest grid point appeared to properly capture
both the sharp drop in SSS values and the persistent low salinity values for the next 4-day
period. By the end of the month, the modelled salinity field seemed to steadily recover to
usual levels in the range of 35.5-35.8 PSU, whereas in situ observations revealed a steeper
rise to 34.8 PSU by the 23[th] of March. Nevertheless, the skill metrics confirmed the accurate
IBI performance, with a correlation coefficient of 0.92 and a RMSD of 0.33 PSU. By contrast,
although GLOBAL outputs could replicate the mean SSS, it did not reproduce satisfactorily
the freshwater episode and barely showed any temporal variability, as reflected by a
negligible correlation coefficient (0.09) and a higher RMSD (0.84 PSU).
Consequently, the impact of colder freshwater river inputs on the SST was also evaluated
(Figure 6, b). Once again, while the sudden cooling of 1.5ºC denoted by in situ observations
was fairly well replicated by IBI, GLOBAL system could only correctly predict the overall
decreasing trend along with the SST values immediately before (13.5ºC) and after (13ºC) the
analysed event. As a consequence, the monthly correlation coefficient (RMSD) obtained for
IBI is higher (lower): 0.79 versus 0.20 (0.25ºC versus 0.35ºC).
The buoyancy input introduced by large freshwaters fluxes (particularly during the spring
freshet), together with topographic effects, contributed to the development of the well-
documented Western Iberian Buoyant Plume (Peliz et al., 2002; Otero et al., 2008), which
strongly influenced the shelf circulation, forming an averaged veering to ~270º (measured
clockwise from the North) during 20[th]-21[st] of March, as reflected by in situ observations and
IBI outputs (Figure 6, c). However, GLOBAL could only partially reproduce the prevailing
surface flow as modelled currents were mainly advected to the south-southwest (180º-270º).
Equally, IBI appeared to correctly replicate the acceleration of the upper-layer stream from 10
to 45 cm·s$^{-1}$ due to impulsive-type freshwater river outflow already observed in situ
estimations of sea surface currents (Figure 6, d). Notwithstanding, GLOBAL current intensity
remained moderated (below 20 cm·s$^{-1}$) during most part of March, including the selected
episode, as reflected by the poorer skill metrics obtained. The current speed underestimation
observed in this tidal environment is mainly attributable to the fact that GLOBAL system
provides a detided solution, so barotropic tidal velocities do not contribute to the final
prescribed total velocity.
Daily-averaged maps of modelled SSS and SST were computed for the 21[st] of March
(Figure 6, e-h) to infer the differences between GLOBAL and IBI. As it can be seen, the
former showed a relatively-smoothed and spatially-homogeneous decrease in the salinity and
temperature fields along the entire coastline (Figure 6, e-f), while the latter exhibited more





intricate patterns with many filaments together with a significant drop in SSS and SST (Figure
6, g-h) in the periphery of the three main local rivers mouth (from North to South: Miño,
Douro and Tagus) as a result of freshwater plumes flowing out over saltier Atlantic waters. In
this three cases, the SST field could effectively act as a tracer for the salinity stratification.
There is a significant resemblance between the monthly current roses derived from in situ
observations and IBI predictions in terms of speed and mean direction (Figure 6, i), showing
the predominance of the so-called Iberian Poleward Current, flowing northwards and
circuiting the western and northern Iberian margins under prevailing southerly winds (Torres
and Barton, 2006). GLOBAL current outputs differed from observations, exhibiting an overall
tendency for eastward directions. The skill metrics derived from time series comparison at B4
buoy location confirmed that the regional OOFS outperformed the global one during March
2018, hence postulating the benefits of improved horizontal resolution to better resolve the
plume dynamics and its extension off-shelf. In addition, the increased horizontal resolution of
IBI allows to better resolving individual frontal fluctuations and horizontal salinity gradients
by preserving the signal of river plume narrower, closer to the coast and with a more complex
structure. The impact of model resolution in both the horizontal extent of the plume and the
strength and position of the freshwater front has been subject of previous studies (Bricheno et
al., 2014). Since both models present 50 depth levels and similar vertical discretization, the
horizontal resolution and the riverine forcing are assumed to play a primary role when
attempting to explain the differences encountered in models performance for this specific test-
case.
**6  Circulation in the Strait of Gibraltar: multi-model inter-comparison from global to**
**coastal scales**
Proved the relevance of the intensity and orientation of the AJ in determining the surface
circulation of the Alboran Sea, the ability of each OOFS to portray the upper layer circulation
in the GIBST area has been evaluated. The annually-averaged surface pattern provided by the
HFR network revealed north-eastward speeds around 100 cm·s$^{-1}$ in the narrowest section of
the Strait (Figure 7, a). SAMPA coastal model seemed to capture well the time-averaged
intensity and orientation of the Atlantic inflow (Figure 7, b), whereas IBI regional model
clearly overestimated the mean surface circulation speed (Figure 7, c). Finally, the coarser
OOFS (GLOBAL) barely captured the most basic features on the incoming flow and its
subsequent propagation towards the north-east (Figure 7, d).
As this qualitative model-intercomparison on a yearly basis was insufficient to infer the
skilfulness of each system, a quantitative validation at the midpoint of the selected transect
(Figure 1, c) was assessed. The scatter plot of HFR-derived hourly current speed versus
direction (taking as reference the North and positive angles clockwise) revealed interesting
details (Figure 8, a): firstly, the AJ flowed predominantly eastwards, forming an angle of 78°
respect the North. The current velocity, on average, was 100 cm·s$^{-1}$ and reached peaks of 250
cm·s$^{-1}$. Speeds below 50 cm·s$^{-1}$ were registered along the entire range of directions.
Westwards currents, albeit minority, were also observed and tended to predominantly form an
angle of 270°.
The scatter plot of SAMPA estimations presented a significant resemblance in terms of
prevailing current velocity and direction (Figure 8, b). Although the time-averaged speed and
angle were slightly smaller (90 cm·s$^{-1}$) and greater (88°), respectively, the main features of the
AJ were qualitatively reproduced: maximum velocities (up to 250 cm·s$^{-1}$) were associated



with an eastward flow and an AJ orientation in the range of 50º - 80º. Besides, surface flow
reversals to the west were properly captured.

3       By contrast, noticeable differences emerged in the scatter plot of regional IBI estimations
(Figure 8, c): surface current velocities below 30 cm·s$^{-1}$ were barely replicated and the AJ
inversion was only observed very occasionally. Despite the fact that IBI appeared to properly
portray the mean characteristics of the eastwards flow, the model tended to privilege flow
directions comprised between 60º and 180º and to overestimate the current velocity, with
averaged and maximum speeds around 117 cm·s$^{-1}$ and 280 cm·s$^{-1}$, respectively.

9       In the case of the scatter plot derived from GLOBAL estimations, even more substantial
discrepancies were detected as the variability of both the AJ direction and speed were clearly
limited to the range 65º-80º and 50-200 cm·s$^{-1}$, respectively (Figure 8, d). No flow reversals
were detected and peak velocities of the eastward flow were underestimated.
The scatter plots of observation-model differences provided relevant information (Figure 8,
e-g). In the case of SAMPA, discrepancies were clustered around zero for both parameters,
with an asymptotic distribution along the main axes (Figure 8, e). On the contrary, a negative
bias to negative differences as observed for both IBI (Figure 8, f) and GLOBAL (Figure 8, g),
especially for the latter. In other words, the regional and global OOFSs overestimated both the
current speed and the angle of the AJ, reflecting a tendency to more south-easterly directions
(clockwise rotated respect the north). Overall, a steady improvement in the AJ
characterization is evidenced in model performance when zooming from global to coastal
configurations, highlighting the benefits of the dynamical downscaling approach.
Additional statistical indicators were computed: two histograms illustrated the number of
hourly zonal (U) and meridional (V) velocity data per class interval (Figure 9, a-b). HFR-
derived zonal velocity estimations exhibited a Gaussian-like shape clustered around 84 cm·s$^{-1}$
and slightly shifted to lower values in the case of SAMPA coastal model (79 cm·s$^{-1}$). Both
datasets show similar positive bias and variability, with the standard deviation around 56-57
cm·s$^{-1}$ for 2017 (Figure 9, a). IBI and GLOBAL presented narrowed histograms, with
distributions positively biased and constrained to zonal velocities above 0 and 40 cm·s$^{-1}$,
respectively. In the case of meridional currents, each distribution exhibits a nearly
symmetrical Gaussian-like shape but biased towards different values (Figure 9, b). Whilst
SAMPA and its parent system IBI exhibited an alike distribution (and moderately similar to
that revealed for HFR estimations), GLOBAL histogram emerged again dramatically
shortened and restricted only to positive values, revealing a recurrent predominance of the AJ
to flow north-eastwards.
Based on the QQ-plot for the zonal velocity component (Figure 9, c), it can be concluded
that SAMPA estimations were consistent despite the slight overestimation observed for the
highest velocities (95$^{th}$–100$^{th}$ percentiles). The general IBI overestimation along the entire
range of percentiles was also clearly evidenced. In accordance with its histogram, GLOBAL
system overestimated (underestimated) zonal currents below (above) the 90$^{th}$ percentile. A
similar behaviour was also observed for GLOBAL meridional velocities, this time around the
20$^{th}$ percentile (Figure 9, d). On the contrary, both SAMPA and IBI appeared to generally
underestimate the meridional surface current speed, even more for higher percentiles.
Class-2 skill metrics, gathered in Table 3, were also computed in order to provide a
quantitative perspective of models performance at the midpoint of the selected transect.
SAMPA clearly outperformed both parent systems, as reflected by lower RMSD values for
both velocity components together with a complex correlation coefficient (CCC) and phase
(CCP) of 0.79 and -8º, respectively, which means that SAMPA predictions were highly



correlated with HFR current observations although slightly clockwise rotated (i.e., more
south-eastwards). The agreement between HFR hourly data and IBI and GLOBAL
estimations, albeit significant (CCC above 0.6), was lower as the related phase values
decreased substantially (especially for GLOBAL: CCP below -20º), indicating a more zonal
surface flow.
The three systems predicted more precisely the zonal velocity component than the
meridional one, with scalar correlations emerging in the ranges [0.68-0.83] and [0.15-0.56],
respectively. Notwithstanding, RMSD were more moderate for the latter (below 37 cm·s$^{-1}$)
than for the former (below 53 cm·s$^{-1}$). This could be attributed to the extremely intense and
predominant West-East zonal exchange of Atlantic-Mediterranean waters through GIBST,
with the meridional flow playing a residual role.
The statistical results derived from SAMPA-HFR comparison, gathered in Table 3, are in
line with those earlier obtained in a 20-month validation performed by Soto-Navarro et al.
(2016), which reported correlations of 0.70 and 0.27 for the zonal and meridional velocities,
respectively. The observed model-radar discrepancies might be attributed to the fact that the
uppermost z-level of SAMPA model is 2.5 m, while HFR observations are representative of
the first 0.5 m of the water column and thus more sensitive to wind forcing. This might
explain some model drawbacks detected in relation to the reduced energy content in surface
current speeds, as reflected by the positive bias between HFR estimations and SAMPA
outputs (Table 3)
Complementarily, the multi-model inter-comparison exercise in the GIBST region focused
on the ability to adequately reproduce an extreme event: the quasi-permanent full reversal of
the AJ surface flow during, at least, 48 hours when intense easterlies episodes were prevalent.
Under this premise, only four episodes were detected and categorized during the entire 2017
(Figure 10). The prevailing synoptic conditions were inferred from ECMWF predictions of
sea level pressure (SLP: Figure 10, a-d) and zonal wind at 10 m height (U-10: Figure 10, e-h).
A significant latitudinal gradient of SLP was observed in 3 episodes (February, March and
December), with high pressures over the Gulf of Biscay and isobars closely spaced in GIBST,
giving rise to very strong easterlies (above 10 m·s$^{-1}$), channelled through the Strait (Figure 10:
e, f and h). In August, the typical summer weather type was observed with Azores High
pressures governing the Atlantic Area and moderate but persistent easterly winds blowing
through the entire Western Mediterranean (Figure 10: c, g).
Both atmospheric variables were spatially-averaged over specific sub-regions (WSMED
and GIBST, respectively, indicated by a red square in Figure 10: a-h) and 3-hourly monitored
along the selected months (Figure 10: i-p). Very high SLP values and extremely high
(negative) U-10 (i.e., intense easterlies) led to a complete inversion of the surface flow, from
the prevailing eastward direction to a westward outflow into the Atlantic Ocean, as reflected
in the Hovmöller diagrams computed for the HFR-derived zonal currents (Figure 10, q-t). In
February, a brief 24-h inversion (related to less intense easterlies) preceded the full reversal of
the surface flow (Figure 10, q). Likewise, the event detected in March consisted of an abrupt
interruption and complete reversal of the eastwards AJ (Figure 10, r). By contrast, in August
and December, the classical AJ inflow into the Mediterranean was only observed in the
southern part of the transect, whereas a weaker coastal counter current was detected flowing
westwards and bordering the Spanish shoreline (Figure 10, s-t). Such coastal flow inversion
has been previously reported and subject to further analysis by Reyes et al. (2015).
Particularly, the flow reversal detected in August was not triggered by high SLP (Figure 10,
k) but induced by moderate and persistent easterlies (5 m·s$^{-1}$, Figure 10-o).





Short-lived reversals of the surface inflow have been previously reported to occur almost
every tidal cycle in Camarinal Sill (western end of GIBST: Figure 1-b) mainly due to the
contribution of the semidiurnal tidal component $M_2$ (Reyes, 2015; Sannino, et al. 2004; García
Lafuente, et al., 1990; La Violette and Lacombe 1988). Since the mean inflow of Atlantic
water is modulated by barotropic tidal currents, hourly-averaged sea surface height (SSH)
observations provided by Tarifa tide-gauge (Figure 1, c) were used to elucidate if the four 2-
day inflow reversal events in the eastern end of the Strait could have been mostly influenced
by spring-neap tidal cycle fluctuations (Figure 10, u-x). Although the fortnightly variability
was clearly observable in a monthly time series of SSH, no cause-effect relationship could be
visually inferred from the inspection of zonal velocities at the selected transect (Figure 10, q-
t). Apparently, evidence of preference for a specific tidal cycle was not observed as the four
flow reversal episodes took place under strong easterlies but during different tidal conditions,
ranging from neap tides (Figure 10, u) to spring tides (Figure 10: v, x). As shown in Lorente
et al. (2018), tides seemed to play a secondary role by partially speeding up or slowing down
the westward currents, depending on the phase of the tide. These results are in accordance with
previous modelling studies (Sannino et al., 2004) where the contribution of the semidiurnal
tidal component to the transport was proved to be relevant over the Camarinal Sill,
(incrementing the mean transport by about 30%, for both the inflow and the outflow), whereas
it was almost negligible at the eastern end of the Strait.
The observed 2-day averaged HFR-derived circulation patterns associated with the four
events here studied were depicted in Figure 11 (a, e, i, m). Some common peculiarities were
exposed, such as the overall westward outflow through the narrowest section of GIBST or the
subtle anticyclonic inflow into the Algeciras Bay. Three study cases revealed a predominant
circulation towards the West together with a marked acceleration of the flow in the periphery
of Algeciras Bay, reaching speeds above 70 cm·s$^{-1}$ (Figure 11: a, e, i). The fourth case
(December 2017) was substantially less energetic and exhibited a rather counter-clockwise
recirculation in the entrance to GIBST. (Figure 11, m). On the other hand, two episodes
illustrated how the circulation in the easternmost region of the study domain followed a
clockwise rotation (Figure 11: e, m).
From a qualitative perspective, SAMPA was able to reproduce fairly well at least two of
the four inversion episodes in terms of overall circulation pattern in GIBST and adjacent
waters (Figure 11: f, j, n). In the event of March, SAMPA replicated the intense eastern
anticyclonic gyre, with velocities up to 80 cm·s$^{-1}$, along with the inflow into the Algeciras
Bay. However, the model could only partially resolve the AJ inversion, exhibiting a counter-
clockwise recirculation with the outflow restricted to the north-western Spanish shoreline
(Figure 11, f). In the episode corresponding to 4$^{th}$–5$^{th}$ of December (Figure 11, n), the upper-
layer dynamic was rather similar to the previously described for March, albeit less vigorous.
The visual resemblance with HFR map (Figure 11, m) was generally high, according to
common features observed: the eastern anticyclonic gyre, the central belt of currents
circulating towards the North-West and eventually the cyclonic recirculation structure in the
entrance to GIBST. On the contrary, in the event occurred between 14$^{th}$-15$^{th}$ of August
(Figure 11, j), a moderate observation-model resemblance was deduced in the northeastern
sector of the domain: SAMPA was able to resolve the observed southwestward stream, the
inflow into the Algeciras Bay and the weak intrusion of Mediterranean waters into GIBST
bordering the northern shoreline but, by contrast, it was ultimately impelled to join the general
AJ inflow governing the Strait and propagating towards the east. Finally, although SAMPA
predicted the occurrence AJ reversal by 20$^{th}$–21$^{st}$ of February (Figure 11, b), the simulated
circulation structure partially differed from that observed with HFR estimations (Figure 11,



a). Whereas the formed prognosticated a meander-like circulation, a predominant cross-shore
stream within the channel and a flow inversion uniquely circumscribed to the entrance of
GIBST, the latter provided an overall westward outflow from the Mediterranean Sea into the
Atlantic Ocean.
In the case of IBI, the Atlantic inflow was always present. In two episodes, the intense AJ
was directed towards the North-East (Figure 11: g, o), converging with the overall clock-wise
gyre that dominated the easternmost region, which was already observed in HFR estimations
(Figure 11: e, m). By contrast, in the two remaining episodes the surface inflow was
predominantly zonal (Figure 11, c) and directed south-eastwards (Figure 11, k), respectively.
Whereas in the former event no common features could be observed between the HFR and
IBI, in the latter a moderate observation-model resemblance was deduced in the northeastern
sector of the domain, as similarly occurred with SAMPA estimations (Figure 11, j). Leaving
aside the counter-clockwise eddy observed in IBI pattern (Figure 11, k), absent from HFR
map (Figure 11, i), IBI partially resolved the observed southwestward flow, the circulation
into the Algeciras Bay and the westward penetration of surface waters along the northern
shoreline of the Strait. Finally, GLOBAL system barely replicated the HFR-derived
circulation patterns as the northeastward stream was permanently locked, showing further
reduced speed variations from one episode to another (Figure 11: d, h, l, p).
Among the physical implications of the surface inflow reversal, abrupt increases in the
SST field were revealed, especially during summertime when warmer surface waters
outflowed into the Atlantic from the Mediterranean (Figure 12). During August 2017, the
aforementioned CCC raised the day $11^{th}$ and lasted until the end of the month, confined at
higher latitudes except for the already analysed 2-day event of $14^{th}$-$15^{th}$, coinciding with the
full reversal mentioned (Figure 12, a). The monthly inter-comparison of the zonal currents at
the midpoint of the selected transect (represented by a black square in Figure 12-a) confirmed
the progressive improvement of the multi-nesting strategy, according to the skill metrics
obtained (Figure 12, b). SAMPA and IBI were able to accurately reproduce the wide tidal
oscillations, although only the former could properly capture the flow inversions represented
by negative zonal velocities that took place between the $14^{th}$-$15^{th}$ and between $21^{st}$-$24^{th}$ of
August. GLOBAL detided outputs only reproduced basic features of the surface flow,
showing always smoothed eastward velocities. As a consequence, skill metrics for the coastal
OOFS were better than for its parent system, and recursively regional skill metrics were in
turn better than global ones, in terms of higher (lower) correlation (RMSD) values. Analysis
for the meridional velocity component (not shown) revealed similar results, with the SAMPA
outperforming the coarser models. Notwithstanding, the three OOFS proved to be more
skilled to forecast zonal than meridional currents. The complex correlation coefficient and the
related phase were 0.85 and -7.37º, respectively, indicating both the relevant SAMPA-HFR
agreement and the slight veering of model outputs respect HFR estimations: a negative value
denoted a clockwise rotation of modelled current vectors (i.e., a more southwardly direction).
In the case of IBI, although the phase was similar (-7.92º) the complex correlation was lower
(0.72). GLOBAL current vectors were, on average, significantly veered clockwise (-25.71º),
despite the high complex correlation coefficient (0.70).
From the $11^{th}$ of August, a progressive warming of 7.5ºC at the upper ocean layer of the
northern shoreline was observed (Figure 12, c), according to the in situ estimations provided
by B7 buoy (whose latitude is located with a solid black dot in Figure 12-a). As easterly
winds progressively dominated the study-area and persisted enough, the CCC broadened and
the complete inflow reversal transported warmer Mediterranean waters to the west through
the entire transect, as reflected by the pronounced SST maximum (~25ºC) detected soon



afterwards, by the 18$^{th}$ of August. A secondary peak of SST was monitored by the 25$^{th}$, before the CCC started weakening. In accordance with previous statements about model behaviour for the zonal currents, once again SAMPA outperformed the parent systems as reflected by a significantly high correlation of 0.89 and a lower but statistically relevant RMSD of 1.22ºC. IBI presented a general bias (positive the first week of august and negative the rest of the month) but adequately reproduced the temporal variability of the SST field (correlation of 0.67). In the case of GLOBAL, the system could not benefit from data assimilation in this intricate coastal area with low level of available observations: worse skill metrics were subsequently obtained, with a correlation of 0.65 and a RMSD above 2ºC.

Finally, outputs from SAMPA high-resolution coastal model were used to provide further insight into the entire AJ-WAG system and how diversity from the classical picture of the Alboran Sea surface circulation emerged from changes in the intensity and direction of the AJ. Although only one episode (corresponding to December 2017) is here shown (Figure 13), the four events followed a similar scenario:

i) *Prelude*: the classical AJ was observed flowing vigorously (with velocities clearly above 80 cm·s$^{-1}$) northeastwards into the Alboran Sea and feeding the WAG (Figure 13, a-b).

ii) *Onset*: as westerly wind lost strength, the AJ speed became progressively weaker and the jet tended to flow more southwardly, giving rise to a weakening and subsequent decoupling of the AJ-WAG system along with the genesis of a new small-scale coastal eddy that coexisted with the WAG (Figure 13, c-d). Circulation snapshots with three gyres (including the EAG, out of the pictures) have been previously reported in the literature (Flexas et al., 2006; Viúdez et al., 1998). The new eddy could be either cyclonic and confined northeast of Algeciras Bay (February 2017, not shown) or anticyclonic and starting to detach from the coast and migrate eastwards (Figure 13, e-f). Meanwhile, the WAG presented different configurations: from an almost-symmetric aspect (August 2017, not shown) to a more elongated shape in the cross-shore direction (December 2017: Figure 13-f) or in the along-shore direction (March 2017, not shown).

iii)*Development*: The AJ velocity reached a minimum (below 50 cm·s$^{-1}$) associated with a sharp change in the predominant wind regime from westerlies to easterlies (Figure 13, g-h). A branch of the eddy, neighboring the Strait, was wind-weakened and deflected from the main rotating pathway and started to flow westwards to the GIBST.

iv)*Full establishment of the inflow reversal*: complete westward outflow from the Mediterranean Sea into the Atlantic Ocean through the narrowest section of GIBST, reaching a peak of velocity over Camarinal Sill (Figure 13, i). The migratory eddy and the WAG started merging into one single anticyclonic gyre (Figure 13, j).

v) *Epilogue*: Afterwards, in three of the cases the re-settlement of predominant westerlies (Figure 10: m, n, p) favoured the return of the northeastward oriented Atlantic inflow and the consequent reactivation of the usual AJ-WAG system (not shown). By contrast, in the fourth episode (August 2017), summer easterly winds kept blowing moderately for two extra weeks (Figure 10, o) but were too weak to preserve the induced reversal, thus the Atlantic inflow reappeared again.

## 7   Conclusions

The current generation of ocean models have undergone meticulous tuning based on several decades of experience. The ever-increasing inventory of operational ocean forecasting



systems provides the society with a significant wealth of valuable information for high-stakes decision-making and coastal management. Some of them are routinely operated on overlapping regions, offering the opportunity to compare them, judge the strengths and weaknesses of each system and eventually evaluate the added-value of high-resolution coastal models respect to coarser parent model solutions.

In this work, a multi-parameter model inter-comparison was conducted at the sea surface, ranging from global to local scales in a two-phase strategy. Firstly, a comparison of CMEMS products (GLOBAL and the nested IBI regional system) was performed against remote-sensed and in situ observations. In terms of temperature, results highlighted the overall benefits of both the GLOBAL data assimilation in open-waters and the increased horizontal resolution of IBI in coastal areas, respectively. IBI outperformed its coarser parent system in those coastal regions characterized by a jagged coastline and a substantial slope bathymetry. As GLOBAL has a smoothed bathymetry and do not resolve many narrow features of the real sea floor, the depths where mixing takes place could be biased. Besides, those mixing processes acting at scales smaller than the grid cell size might substantially affect the resolved large-scale flow in the coarser GLOBAL system.

On the other hand, since GLOBAL is a detided model solution, tidally-driven mixing could account for a portion of the discrepancies found between GLOBAL and satellite-derived SST estimations in energetic tidal areas such as the English Channel, the North Sea and the Irish Sea. Whereas GLOBAL seemed to predict an over-stratification in shelf-seas, IBI could better reproduce the vertical stratification and hence the SST field in the aforementioned subregions.

Complementarily, an isolated but rather illustrative example of the impact of impulsive-type river freshwater discharge on local surface circulation in NW Spain was provided. The increased horizontal resolution of IBI allowed a more accurate representation of horizontal salinity gradients, the horizontal extent of the plume and the strength and position of the freshwater front, according to the results derived from the validation against in situ observations of SSS, SST and currents provided by a moored buoy. Since both GLOBAL and IBI present 50 depth levels, similar vertical discretization and comparable climatological runoff forcing, the horizontal resolution is assumed to play a primary role when attempting to explain the differences encountered in models performance for this specific test-case. Notwithstanding, the authors are fully aware of this single isolated example does not suffice and additional events over the entire IBI coastal domain should be examined in future works.

Finally, a 1-year (2017) multi-model inter-comparison exercise was performed in the Strait of Gibraltar between GLOBAL, IBI and the nested SAMPA coastal system in order to elucidate the accuracy of each OOFS to characterize the AJ dynamic. A quantitative comparison against hourly HFR estimations highlighted both the steady improvement in AJ representation when moving from global to coastal scales though a multi-nesting model approach and also the relevance of a variety of factors at local scales, among others:

i) A sufficiently detailed representation of bathymetric features: the very high horizontal resolution of SAMPA (~ 400 m) and, consequently, the tailored bathymetry employed in order to capture small-scale ocean process and resolve sharp topographic details.

ii) A better representation of air-sea interactions: the adequate refinement of the spatio-temporal resolution of the atmospheric forcing used in SAMPA, especially in a complex coastal region where topographical steering further impacts on flows.

iii) The inclusion of accurate tidal and meteorologically-driven barotropic velocities, prescribed across the open boundaries, allowed a detailed examination of persistent



Atlantic inflow reversal episodes. Although the matching between HFR observations
and SAMPA outputs is mainly found in two of the four reversal events detected, this
result demonstrates its added value as modelling tool towards the comprehension of
such singular oceanographic event. A detailed characterization of this phenomenon is
relevant from diverse aspects, encompassing search and rescue operations (to
adequately expand westwards the search area), the management of accidental marine
pollution episodes (to establish alternative contingency plans), or safe ship routing (to
maximize fuel efficiency).
Finally, SAMPA coastal model outputs were analysed in order to put in a broader
perspective the context of the onset, development and end of such flow reversal and its impact
on the AJ-WAG coupled system. The synergistic approach based on the integration of HFR
observing network and SAMPA predictive model has proved to be valid to comprehensively
characterize the highly dynamic coastal circulation in the GIBST and the aforementioned
episodic full reversals of the surface inflow. In this context, data assimilation would provide
the integrative framework for maximizing the joint utility of HFR-derived observations and
coastal circulation models. A data assimilation scheme could be incorporated in future
operational versions of SAMPA in order to improve its predictive skills, since similar
initiatives are currently ongoing with positive results (Hernández-Lasheras et al., 2018;
Vandenbulcke et al., 2017; Stanev et al., 2015).
Future efforts are planned to improve CMEMS global and regional OOFSs in several
aspects already addressed in the present work. While GLOBAL system will be evolved
towards a 1/36º model application, a substantial refinement will be accomplished for regional
IBI system in both vertical and horizontal resolutions: from 50 to 75 depth layers and from
1/36º to 1/108º (~1 km), respectively. Whereas the first feature will be incorporated during
CMEMS Phase-2 (2018-2021), the second milestone will be achieved in the frame of Inmerse
H2020 project and is expected to positively impact on a more accurate representation of
coastal processes, among others: submesoscale shelf break exchanges and connectivity,
fronts, river plumes or topographic controls on circulation.
In addition, a more detailed bathymetry is expected to be introduced in future operational
versions of IBI in order to better resolve those regions with complex coastline and intricate
bottom topography. Other factors that could be potentially improved but still deserve further
analysis are the air-sea and the land-sea interactions, i.e., the meteorological and riverine
forcings. With regards to the former, a more skilful atmospheric forecast model with a higher
spatiotemporal resolution (i.e., hourly prediction over a more refined grid) could aid to better
represent the coastal circulation by a more accurate discrimination of the topographic
structures and the replication of the inertial oscillations and mesoscale processes. On the other
hand, each main river basin hydrology should be more accurately represented with daily-
updated outputs from tailored hydrological models. Finally, refined mixing schemes might
also produce notable improvement in the representation of water masses, resulting in a
substantial reduction of temperature and salinity bias relative to model solution.




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



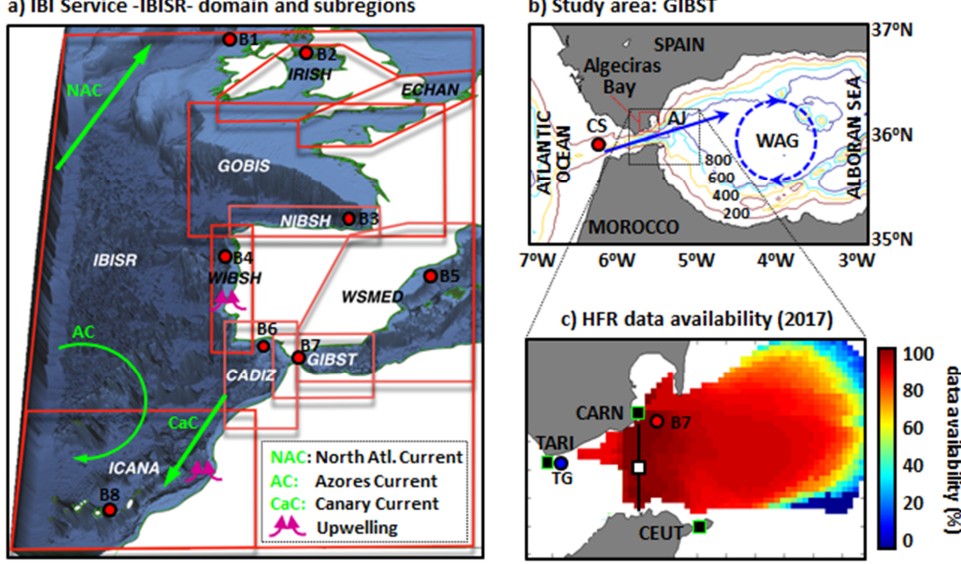

Figure 1. a) Iberia-Biscay-Ireland Service (IBISR) domain, which comprises 9 sub-regions denoted by red squares. Red filled dots represent buoys locations. b) Study area 2: surface Atlantic Jet (AJ) flowing through the Strait of Gibraltar into the Alboran Sea, feeding the Western Alboran Gyre (WAG); isobath depths are labeled every 200 m. Red dot indicates a topographic feature: Camarinal Sill (CS). c) HFR hourly data availability for 2017: solid black squares represent radar sites, blue and red dot indicate Tarifa tide-gauge and B7 buoy location, respectively.





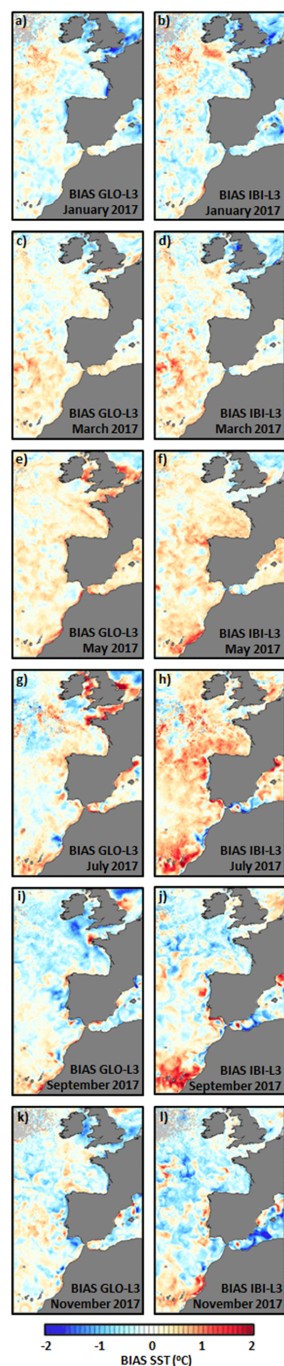

2   Figure 2. Monthly SST bias (model minus observation), where the satellite-derived daily data
3   used is L3 CMEMS operational product: GLOBAL versus observation (left) and IBI versus
4   observation (right).



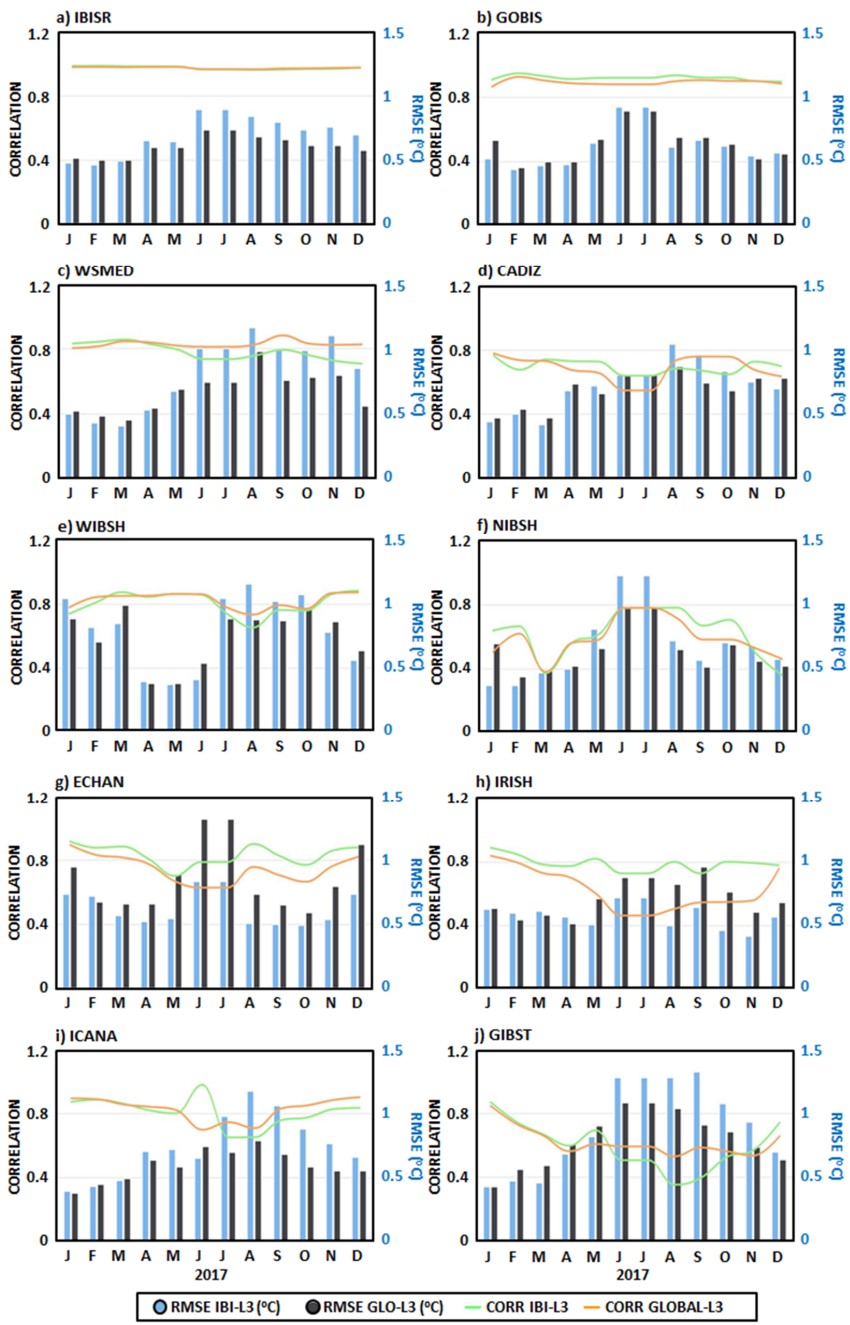

Figure 3. Annual evolution (2017) of monthly skill metrics derived from the comparison of
GLOBAL and IBI models against satellite-derived observations (L3) over IBI service domain
(IBISR) and nine sub-regions, denoted in Figure 1-a. RMSD values and correlation
coefficients are represented by columns and lines, respectively.




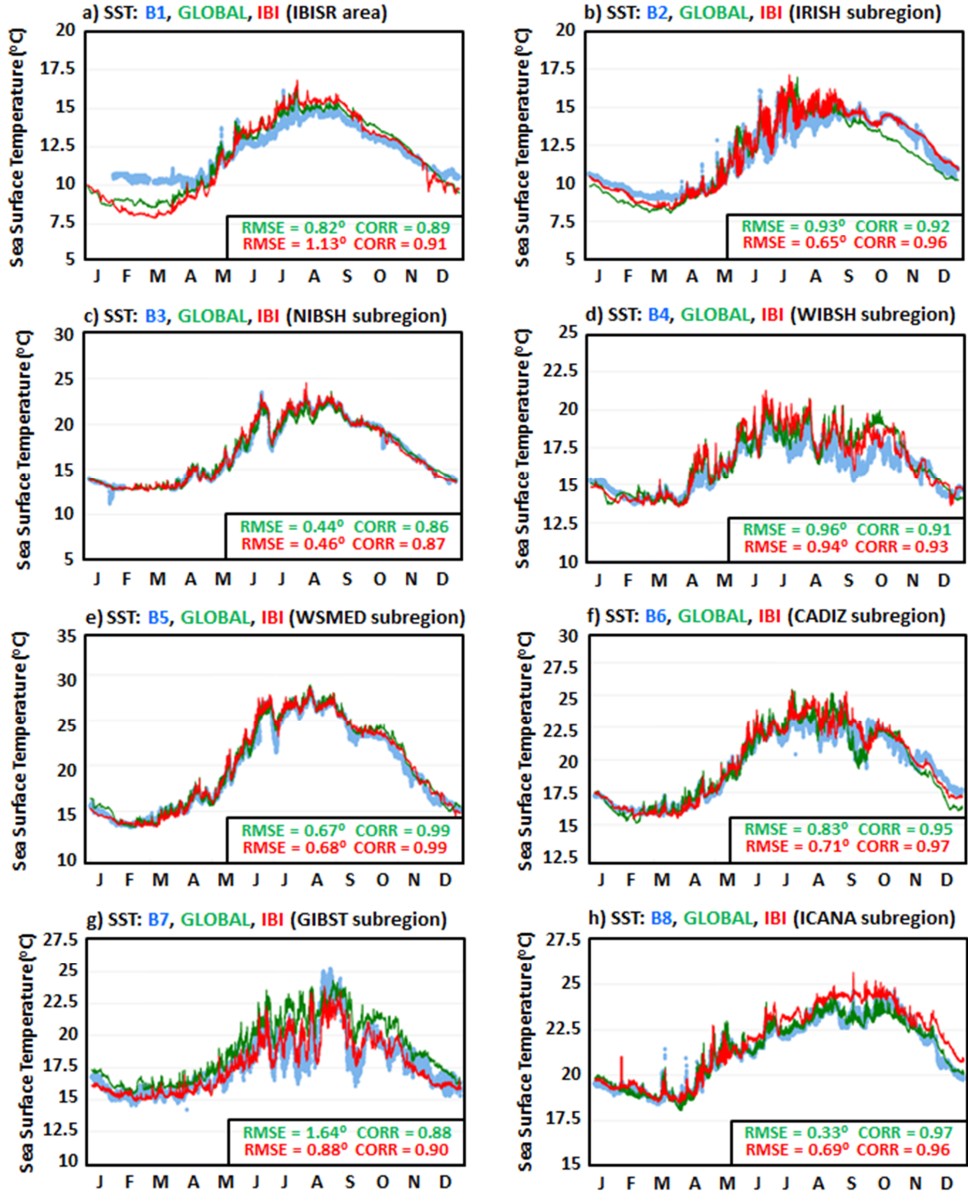

Figure 4. Annual (2017) time series of hourly Sea Surface Temperature (SST) at eight different locations within IBISR area. In situ observations from moored buoys (blue dots), GLOBAL model predictions (green line) and IBI model outputs (red line) are depicted. Skill metrics derived from model-observation comparison are gathered in black boxes.





Figure 5. Annual evolution (2017) of seasonal skill metrics derived from the comparison of
GLOBAL and IBI models against in situ SST hourly observations provided by eight buoys.
RMSD and correlation coefficient represented by coloured bars and lines, respectively.





Figure 6. (a-d) Monthly inter-comparison (March 2018) between GLOBAL (green line), IBI (red line) and B4 buoy (blue dots): sea surface salinity (SSS), temperature (SST), current direction (SCD) and current speed (SCS); (e-f) Daily maps of SSS and SST derived from GLOBAL outputs for the 21st of March. Red filled dot represents B4 buoy location; (g-h) Daily maps of SSS and SST derived from IBI outputs for the 21st of March; (i) Monthly surface current roses. Monthly skill metrics derived from model-observation comparisons are provided.



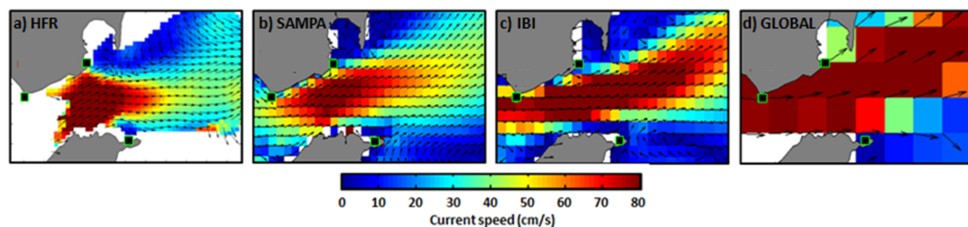

3    Figure 7. Annual mean circulation pattern in GIBST for 2017, derived from hourly

4    estimations provided by: a) HFR; b) SAMPA coastal model; c) IBI regional model; d)

5    GLOBAL model. For the sake of clarity, only one vector every two was plotted in HFR map.



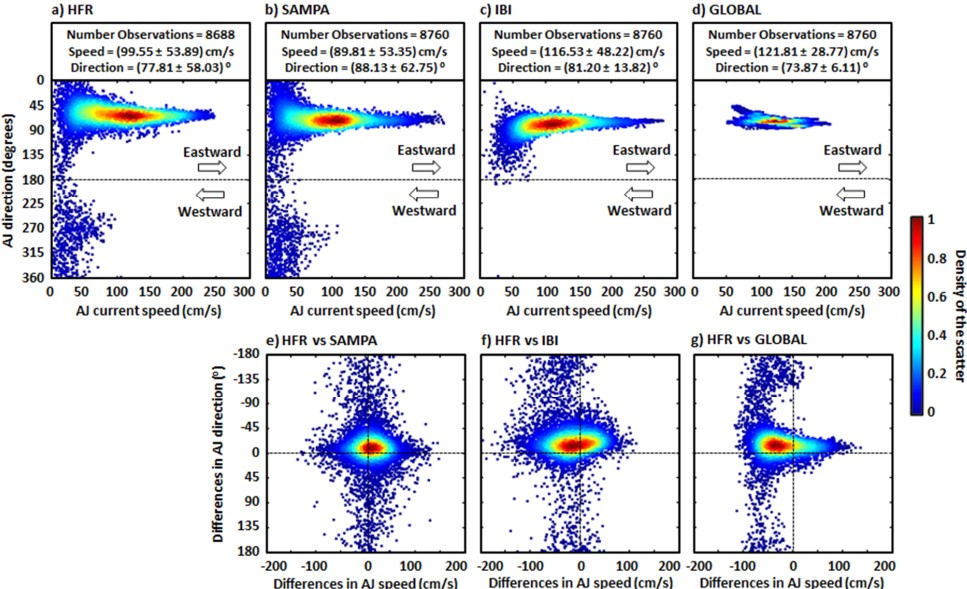

Figure 8. (a-d) Annual (2017) scatter plot of hourly AJ current speed versus direction (angle
measured clockwise from the North); estimations provided by: a) HFR; b) SAMPA; c) IBI; d)
GLOBAL. Mean and standard deviation values of both AJ speed and direction are gathered in
black boxes; (e-g) Annual scatter plot of differences (observation minus model) in AJ speed
and direction between: e) HFR and SAMPA; f) HFR and IBI; g) HFR and GLOBAL.



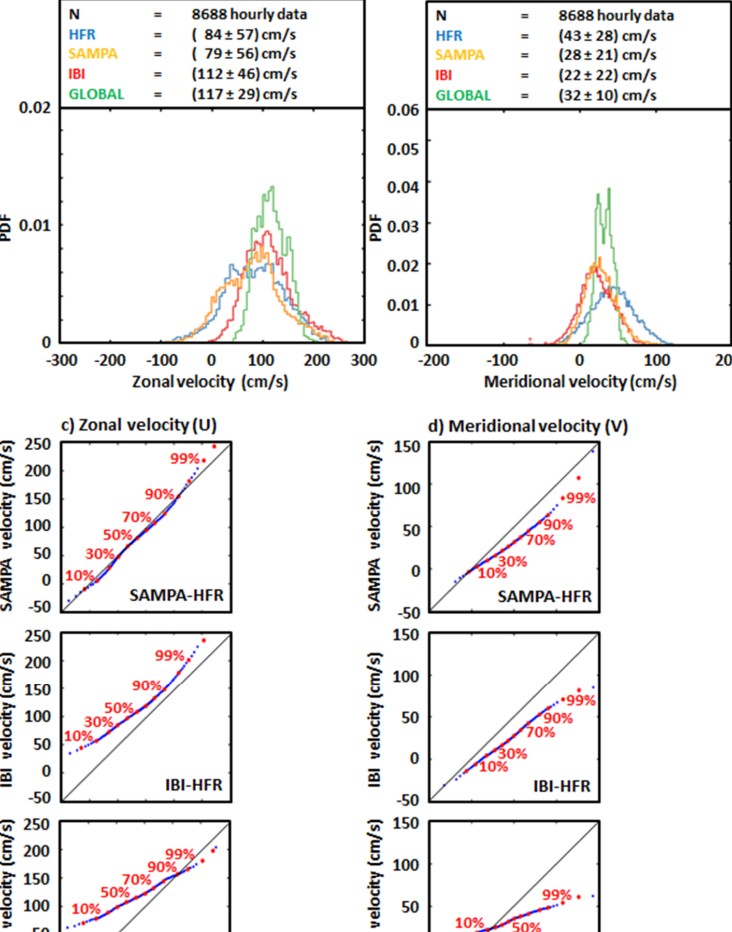

Figure 9. Annual (2017) histogram of hourly: (a) zonal current velocities; (b) meridional
current velocities, as provided by HFR, SAMPA, IBI and GLOBAL. Mean and standard
deviation values are gathered in black boxes. Quantile-quantile plots of hourly: (c) zonal
current velocities; (d) meridional current velocities, as derived from the observation-model
comparison. 10–99% quantiles were established (red filled dots);

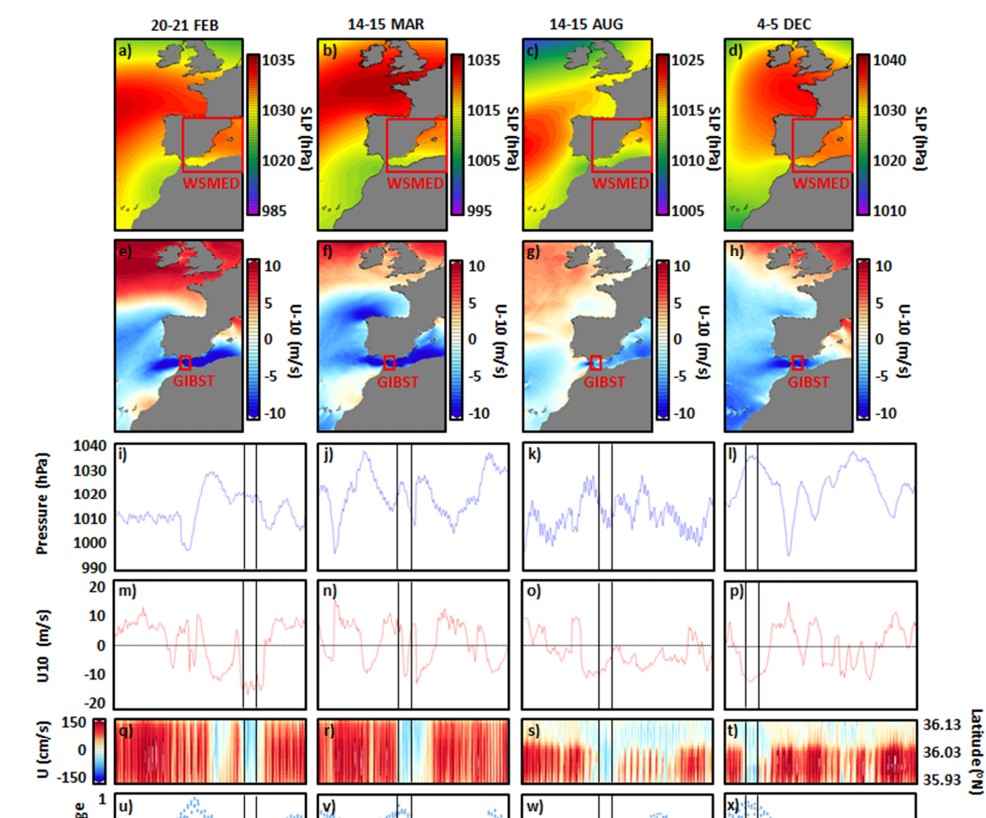

Figure 10. 2-day averaged synoptic maps of: (a-d) sea level pressure (SLP); (e-h) zonal wind
at 10 m height (U-10), provided by ECMWF, corresponding to each of the four Atlantic
inflow reversal events analysed during 2017. (i-l) Monthly time series of SLP, spatially
averaged over the Western Mediterranean (WSMED) subregion, marked with a big red box in
the maps of the first row; (m-p) Monthly time series of U-10, spatially averaged over the
Strait of Gibraltar (GIBST) subregion, marked with a small red box in the maps of the second
row; (q-t) Monthly Hovmöller diagrams of HFR-derived zonal current velocity at the selected
transect. Red (blue) colour represent eastward (westward) flow; (u-x) Monthly time series of
hourly sea surface height (SSH) provided by Tarifa tide-gauge, represented by a blue dot in
Figure 1-c. 2-day episodes of permanent flow reversal are marked with black boxes in (i-x).




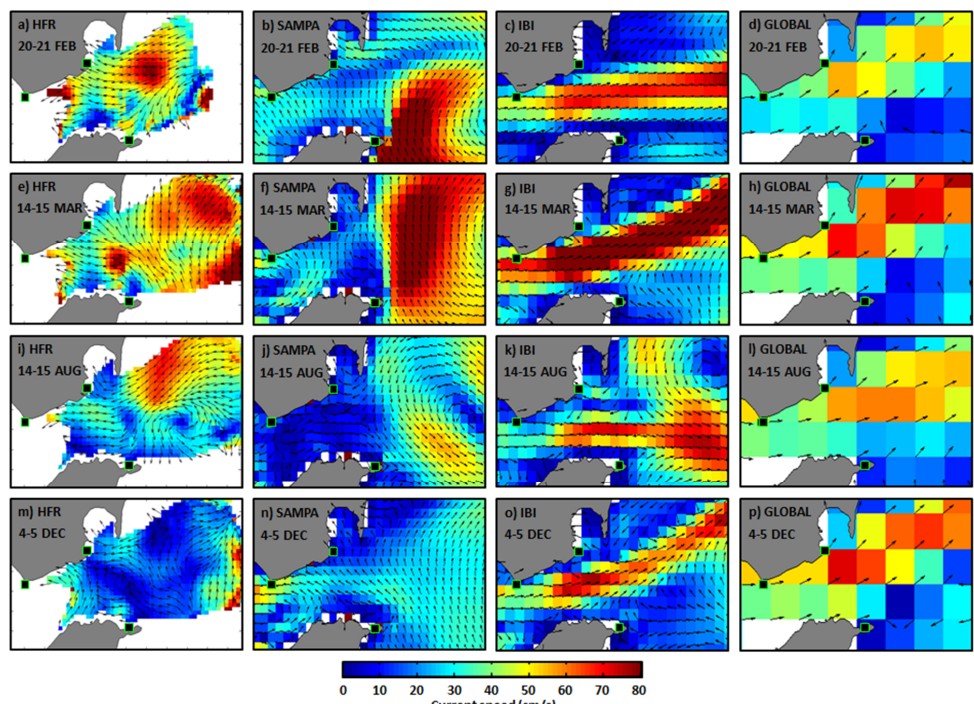

Figure 11. 2-day averaged maps of the surface circulation in GIBST, corresponding to each of
the four Atlantic inflow reversal events detected in 2017 (from top to bottom). Maps derived
from hourly estimations were provided by (from left to right): HFR, SAMPA coastal model,
IBI regional model and GLOBAL model. For the sake of clarity, only one vector every two
was plotted in HFR map.



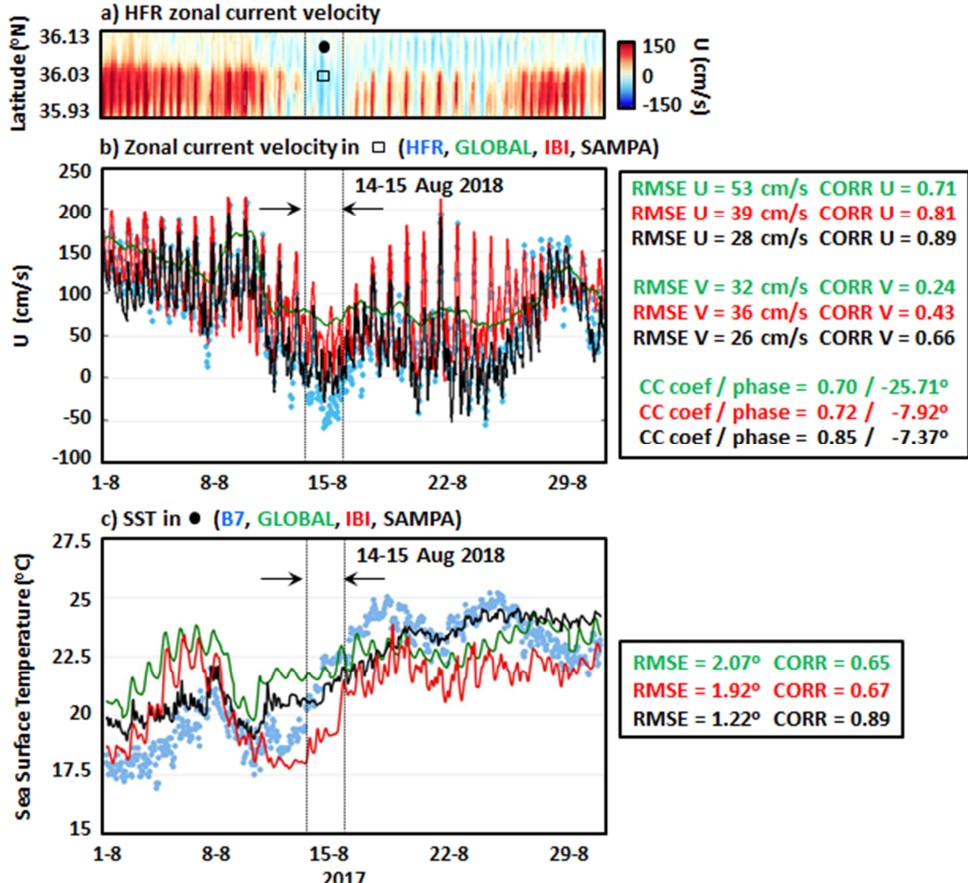

Figure 12. a) Monthly Hovmöller diagram of HFR-derived zonal current velocity at the
selected transect in the Strait of Gibraltar for August 2017. Red (blue) colour represent
eastward (westward) flow. A complete Atlantic inflow reversal episode marked with black
box for the 14-15 August; b) Monthly times eries of zonal current velocity at the midpoint of
the transect (represented by a black square in the Hovmöller diagram) provided by HFR (blue
dots), SAMPA (black line), IBI (red line) and GLOBAL (green line); c) Monthly time series
of SST at B7 buoy location (represented by a solid black dot in the Hovmöller diagram)
provided by B7 buoy (blue dots), SAMPA (black line), IBI (red line) and GLOBAL (green
line). Monthly skill metrics derived from observation-model comparison are gathered in black
boxes on the right.

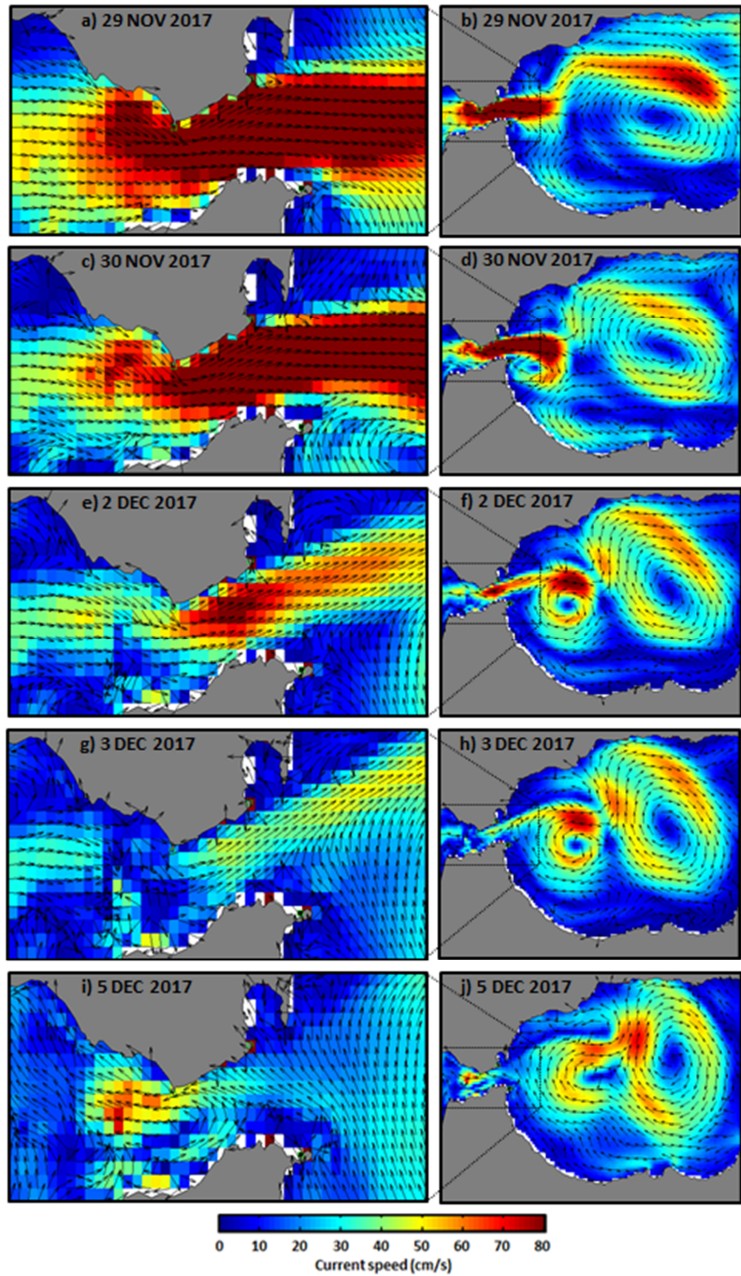

Figure 13. Sequence of SAMPA daily surface circulation maps covering the period from the 29[th] of November to the 5[th] of December 2017. General map on the right and zoom over the Strait of Gibraltar on the left. An inflow reversal through the narrowest section of the Strait of Gibraltar is evidenced by the 5[th] of December, as a result of a change in the wind regime, from westerlies to easterlies.



| Features \ Model | CMEMS GLOBAL | CMEMS IBI | SAMPA |
|---|---|---|---|
| Model | NEMO 3.1 | NEMO 3.6 | MITgcm |
| Configuration | Global | Regional | Coastal |
| Domain: lon, lat | 180ºW-180ºE, 89ºS-90ºN | 19ºW-5ºE, 26ºN-56ºN | 7.4ºW-3ºW, 35ºN-37.2ºN |
| Resolution | 1/12º | 1/36º | Variable (300-500 m at GIBST) |
| Product grid points | 4320 x 2041 | 865 x 1081 | 200 x 100 |
| Forecast (days) | 10 | 5 | 3 |
| Forecast update | Daily | Daily | Daily |
| Depth levels | 50 (unevenly distributed) | 50 (unevenly distributed) | 46 (unevenly distributed) |
| Initial conditions | EN4 climatology | GLOBAL | IBI + NIVMAR |
| Open boundary conditions | NO | Daily 3D data from CMEMS GLOBAL | Daily 3D data from CMEMS IBI + barotropic velocity from NIVMAR |
| Atmospheric forcing | ECMWF (3-h) | ECMWF (3-h) | AEMET (1-h) |
| Rivers forcing | Monthly climatology | Climatology + Previmer + SMHI | NO |
| Tidal forcing | NO | 11 tidal harmonics from FES2004 and TPXO7.1 models | 8 tidal harmonics from FES2004 (MOG2D model) |
| Assimilation | YES (SAM2) | NO* | NO |
| Bathymetry | ETOPO1 + GEBCO8 | ETOPO1 + GEBCO8 | IOC + high resolution charts |

2  Table 1. Basic features of the ocean forecast systems employed in the present study. * The
3  operational version of IBI here used with spectral nudging. Assimilation scheme SAM2 was
4  later  introduced in v4 (April 2018).



| Buoy | Model | Year | Location: lon, lat | Subregion | Depth (m) | Sampling |
|---|---|---|---|---|---|---|
| **B1** | WaveScan | 2008 | 9.07ºW, 54.67ºN | IBISR | 72 | 1 h |
| **B2** | WaveScan | 2008 | 5.42ºW, 53.47ºN | IRISH | 95 | 1 h |
| **B3** | SeaWatch | 1990 | 3.09ºW, 43.64ºN | NIBSH | 870 | 1 h |
| **B4** | SeaWatch | 1998 | 9.43ºW, 42.12ºN | WIBSH | 600 | 1 h |
| **B5** | SeaWatch | 2004 | 1.47ºE, 40.68ºN | WSMED | 688 | 1 h |
| **B6** | SeaWatch | 1996 | 6.96ºW, 36.48ºN | CADIZ | 450 | 1 h |
| **B7** | WatchKeeper | 2010 | 5.42ºW, 36.07ºN | GIBST | 40 | 1 h |
| **B8** | Triaxys | 1992 | 15.39ºW, 28.05ºN | ICANA | 30 | 1 h |

2  Table 2. Description of the network of directional buoys used in this work. Year label stands
3  for year of deployment. Subregions are defined in Figure 1-a.

| Metrics \ HFR vs: | GLOBAL | IBI | SAMPA |
|---|---|---|---|
| **Bias U (cm·s⁻¹)** | -32.98 | -28.25 | 4.17 |
| **RMSD U (cm·s⁻¹)** | 52.89 | 50.89 | 33.58 |
| **CORR U** | 0.71 | 0.68 | 0.83 |
| **Slope U** | 0.37 | 0.55 | 0.82 |
| **Intercept U (cm·s⁻¹)** | 85.93 | 65.46 | 10.77 |
| **Bias V (cm/s)** | 10.52 | 20.32 | 15.19 |
| **RMSD V (cm·s⁻¹)** | 30.57 | 36.09 | 28.48 |
| **CORR V** | 0.15 | 0.33 | 0.56 |
| **Slope V** | 0.05 | 0.26 | 0.41 |
| **Intercept V (cm·s⁻¹)** | 29.98 | 11.27 | 10.17 |
| **Complex CORR** | 0.67 | 0.62 | 0.79 |
| **Phase (º)** | -22.72 | -12.68 | -7.86 |

12  Table 3. Skill metrics derived from the 1-year (2017) validation of sea surface currents
13  estimated by three operational forecasting systems against HFR-derived observations at the
14  midpoint of the selected transect in the Strait of Gibraltar (Figure 1, c).

