# Peer review of "Skill assessment of global, regional and coastal circulation forecast models"

_Ocean Science, 2018_

## Referee Comment (RC1) · Diego Macias (Referee) · 18 Feb 2019

Review of manuscript # OS-2018-168

In this paper the authors perform a multi-model comparison in the regions surrounding the Iberian Peninsula. From global models using data-assimilation to local models nested into larger scale ones, the authors explore the differences in surface ocean properties for each approach and provides hypothesis and reasoning for the observed patterns. The paper is in general well written with only a few grammatical errors (see

details below) and easily understandable.

Although I appreciate the approach and the effort to objectively analyze pros and cons of the different models I do have some concerns on the present version of the manuscript. Hopefully, such concerns could be solved through a revision of the text so the manuscript could be made acceptable for publication.

Major concerns:

My first issue could be derived from my own lack of expertise with data-assimilation models but I do find it difficult to completely understand how the CMEMS global model works. It is stated (page 6, line 5) that the system provides 10-days forecasts updated daily. Does this mean that every-day the system assimilate all available information to update its status and then is run for 10 days? Then, the next day the cycle re-start, assimilating data for the new day and re-running the system for another 10 days? If I understand this correctly, the Global model is only left 'free' for one day at a time, am I right?

If the above is correct I wonder how it is possible for the model to present such relatively large deviations with respect to satellite in terms of SST (bias range -2/+2, figure 2). This is particularly shocking for me as satellite SST is part of the data assimilated by the model (as you state in page 6, line 26). I understand the calibration/validation is not a task to be performed by the authors but I would like to know your opinion about the system operation, do you have any thoughts on how to improve this issues? Or maybe I totally miss-understood how the system works?

My second issue comes from your interpretation of the results in the Strait of Gibraltar. The improvement in AJ direction and speed from the global to the regional model is clear, however the reason for such are not that obvious as you seem to propose. I fully agree with you that increasing spatial resolution (global<IBI<SAMPA) is one of the major reason why the direction and speed of the AJ is better reproduced in the regional model. However, the inversion events could not be related with this issue. In fact, a

model with similar resolution to Global (see Macias et al., 2016) was able to reproduce the inversions of the jet. In that work, the remote barotropic effect of the meterological forcing over the Mediterranean Sea was proposed as one of the major players in the regulation of the seasonal cycle of the AJ and of its occasional inversions. As far as I understood, only SAMPA include this type of effects (page 7, lines 43-48) and, in my opinion, this is the main reason why only this model is able to correctly reproduce the inversion events. I would suggest to make this difference clear in the text; increasing resolution helps with the simulation of direction/velocity of the jet; correct atmospheric forcing (remote) is essential to get the flow inversions

I strongly suggest the authors to explore these caveats and to try to address them in a revised version of the manuscript.

Minor details:

Page 2, lines 12-17: I don't think global models are able to 'properly resolve' biogeo-chemical cycles, not even at large scale. Also, this phrase is too long, please consider breaking it up. Page 3, line 24: consider changing 'lower' with 'less' Page 3, line 36: what does 'poorly controlled information' exactly means? Page 3, line 44: 'researchers' should be 'research' Page 4, line 28: please indicate in caption of Fig. 1c what the white square represents Page 6, section 3.1: as indicated above, I don't fully understand how the assimilation/run/re-start cycle of this model works. Could you please provide a more detailed explanation? Page 6, section 3.2: similarly, the transfer of information from the Global to the IBI system is not fully clear. Does IBI have some data assimila-tion scheme? Or only information from the parent system is transferred into the model domain? How often the nudging is done? Page 9, lines 32 and 33: the symbol '∘' is missing Page 10, line 29: the transect used for evaluation is the black line/white square in Figure 1c? Page 11, line 15: I can't see any clear benefit in using the IBI over the Global model in here.. Page 11, line 23: why the advantages of the SST assimilation into the Global model is not 'propagated' into the IBI? Is it related with the frequency of the nudging? The method? Page 11, lines 43-47: this explanation does not seems fully

justified. The SST anomalies (even in IBI) does not only occur in the coasts, but also many km away where satellite images should not have any issues. Page 12, line 2: could you please explain better how the correlation spatial maps are computed? Page 12, line 12: what do you mean with 'in like fashion'? Page 12, line 19: it is curious that high 'r' are coincident with high 'RMSD' Page 12, lines 29 – 37: as you are mentioning this 3D comparison some numbers (statistics) should be provided (no figures might be needed though) Page 12, line 45: the point-wise comparisons you provide in Figure 4 seems to have lower biases than most of the maps shown above.. isn't it a bit strange? Page 13, lines 10-12: how an intrusion of warmer waters could make the SST to drop? Page 13, line 27: to avoid this bias you could just extract model data from the closet depth to the buoys? Or make an interpolation to the specific depths? Page 13, lines 37 – 38: I am left wondering if NEMO vertical structure (stability) could be partially responsible for the observed differences. As mentioned above, a data-driven model running freely only for a very limited time should not show such large biases in SST. I know for a fact that NEMO has difficulties to simulate the vertical structure of the water column in the Mediterranean Sea and was wondering if something similar could be happening elsewhere? Page 14, line 18: 'accurate' seems a bit subjective.. why not use 'rather accurate' instead? Page 14, line 24: the cooling in the IBI simulation does not only occur along the river plume but also on the NW Iberian coast. Could it be also related with some other process happening a more regional scale? Such as locally-induced upwellings? Page 16, line 34: red line in Figure 6c is quite difficult to interpret because of the continuous changes associated to the tidal cycle. If you use dots (as with the buoy data) it might be more easy to read the figure. Page 15, line 4: it is true that SST decrease on the river plume but, as commented above, in IBI there are other processes bringing up cold waters nearby the coast. Page 16, line 21: the differences between the different models are not just on the downscaling (increasing resolution) but also on the imposition of lateral conditions at the boundaries! Page 16, line 26: positive bias with respect what? Page 17, line 44: the two ways current system you describe in here is not clear from the graphs in Figure 10 Page 19, line

27: the fact that SAMPA and IBI represents the tidal dynamics is not because of the nesting, but because you include this forcing in both models (and not in Global) Page 19, line 35: increasing resolution is not the only reason why SAMPA outperforms the other two models (see general comment above) Page 19, lines 36 -42: where are the metrics you refer here to? Page 19, line 43: the warming in Fig. 12c is less than 7.5 degrees, I would say ~5? Page 20, line 15: the situation of the WAG you describe here is not the typical one. The AJ is entering in a rather meridional direction and the WAG seems to be slightly detached from the NW Alboran. I would say this is already an evolving situation into the inversion episode Page 20, line 20: the coastal eddy in the NW Alboran is almost always there (see situation in previous snapshot and plenty of reports elsewhere), the only difference is how big this structure is (which is linked to the AJ migration and WAG displacement) Page 21, point iii): you acknowledege here the potential effects of barotrophic flows on AJ inversions but is not clear in your discussion above. As suggested in the general comments, I would recommend to make a stronger case for this difference between models, resolution is important for the Strait dyanmics but is not the only element to consider Page 22, lines 14 – 19: I would also recommend to keep on improving the mechanics of the models. Data assimilation is a nice tool but should be developed on parallel with model improvements. Otherwise models would only become very sophisticated data-interpolation tools, losing their potential to fill gaps by doing free-simulations. Page 22, lines 38 – 39: see my comment above about NEMO and vertical stability problems in the Mediterranean. We should think about which model is best to be used depending on its applications

---

## Referee Comment (RC2) · Christian Ferrarin (Referee) · 21 Mar 2019

In this manuscript, titled "Skill assessment of global, regional and coastal circulation forecast models: evaluating the benefits of dynamical downscaling in IBI surface waters", the authors describe an inter-comparison of different oceanographic operational systems in the Iberian shelf and in the Gibraltar Strait. Although the numerical results are deeply investigated showing the different model performance in reproducing surface water properties, to my opinion the manuscript suffers from a lack of identification of the novel aspects of the research. Al stated by the authors, the three considered

operational systems have been already described in many publications. Similarly, the benefits of downscaling (i.e. increasing resolution) in coastal waters have been widely treated in literature.

Major comments:

1. I appreciate the authors for highlighting the need of accurately addresses land-sea, air-sea, and coastal-offshore interactions when dealing with regional operational ocean systems (see also Kourafalou et al. 2015; Wilkin et al. 2017; Ferrarin et al., 2019). However, I think the authors should mention in the introduction that the downscaling from open sea (ocean) to coastal area can be also achieved through the implementation of numerical models based on a unique unstructured grid (Cucco et al. 2012; Ferrarin et al. 2013, 2019; Zhang et al. 2016; Federico et al. 2017; Stanev et al. 2017) able to describe processes at different spatial scales, and not only or through nesting of models. I'd like to point out that unstructured approach is particularly useful for realising a seamless transition between adjoining regional seas connected by narrow straits (i.e. the Gibraltar Strait; Stanev et al. 2017 and Ferrarin et al., 2018).

2. The only novel aspect in this work seams to be the inclusion of the so called "spectral" nudging technique into the operational IBI system. A similar methodology has been recently applied by Ferrarin et al. (2019) in the Tiresias operational system with a relaxation coefficient spatially varying over the Adriatic Sea domain (as a function of the grid resolution) from 2 days in the open sea and increasing, thus diminishing the restoration contribution, toward the coast. The authors should provide more details about the nudging methodology in terms of variables assimilated and 3D spatial weight functions (relaxation time). Moreover, it is not clear what are the benefit of this improvement, since according to the results presented in Fig 2-5, IBI performance is lower to the GLOBAL model on open sea regions (IBISR, ICANA, WSMED), where ideally nudging allows the model state to be reconciled with the assimilated GLOBAL data. Do the differences with the GLOBAL demonstrate that over the sea surface the impact of the air-sea heat fluxes is stronger that the restoration contribution of the GLOBAL

nudging? In this contest, the authors should demonstrate the benefit of the nudging approach analysing the water column properties and not only the sea surface.

3. One of the most important topic described in the manuscript is the model comparison in the Gibraltar Strait. As one would expect, the SAMPA high resolution model performs better in reproducing hydrodynamics in this region having complex morphology and dynamics. However, it is not clear to me if this improvement is only due to the model spatial resolution or to the inclusion of the remote effect of the atmospheric forcing in the barotropic flow thought the strait. According to the model description reported at page 7, SAMPA is not directly nested in IBI since it is forced by daily fields (from IBI) and tidal and meteorological-driver barotropic velocities from MOG2D‐G and NIVAR. In this contest the authors should:

3.1. make clear throughout the whole manuscript that SAMPA is not only nested in IBI. Therefore, IBI could not be defined the only parent system of SAMPA.

3.2. explain the reason of this choice, which sound counter-intuitive to me, since IBI already account for tide and meteorological forcing.

3.3. the authors should discuss the results of the inter-model comparison in the light of the different approaches adopted for the forcing and parametrization in the three systems, not only resolution.

Minor comments:

Pg. 2, line 35: Is Herbert et al., 2014 (on Mercator Ocean - Quarterly Newsletter) a peer review publications? Only peer review articles should be cited.

Pg. 3, line 20: Sotillo et al., 2018. Do not include under review article.

Pg. 3, lines 22-29: The part regarding operational products for harbours seams out of topic.

Pg. 5, line 8: Please provide a reference for this statement.

Pg. 6, lines 10-24: Provide more details about the nudging approach.

Pg. 8, lines 13-14 and 38-41: see major comment 3.

Pg. 10: Section Temperature should be shortened.

Pg. 12, line 6: "GLOBAL performance was more accurate from July April to December as indicated by the lower RMSD".

Pg. 12, line 28-29: This is not true, since divergence between IBI and GLOBAL forecast are found also in the open sea.

Pg. 16, line 44: "... midpoint of the selected transect (white square in Fig. 1c)."

Reference: authours should limit self referencing (29/76) and include only peer reviewed articles without considering under review works (Sotillo el al., 2018), conference abstract (Hernández-Lasheras et al., 2018) or poster (Lorente et al., 2017).

Pg. 24, line 19: provide doi for "in press" article.

Pg. 24, line 32: provide the full list of authors.

Pg. 26, line 28: the journal for this reference is missing.

Figure 1: please indicate the SAMPA model domain; specify the meaning of the black line and white square in panel C. Table 2: Open boundary conditions for SAMPA should also include MOG2D‐G.

Suggested refreferences:

Cucco A, Sinerchia M, Ribotti A, Olita A, Fazioli L, Perilli A, Sorgente B, Borghini M, Schroeder K, Sorgente R. 2012. A high-resolution real-time forecasting system for predicting the fate of oil spills in the Strait of Bonifacio (western Mediterranean Sea). Mar Pollut Bull. 64(6):1186–1200.

Federico I, Pinardi N, Coppini G, Oddo P, Lecci R, Mossa M. 2017. Coastal ocean forecasting with an unstructured grid model in the southern Adriatic and northern Ionian

seas. Nat Hazards Earth Syst Sci. 17(1):45–59.

Ferrarin C, Roland A, Bajo M, Umgiesser G, Cucco A, Davolio S, Buzzi A, Malguzzi P, Drofa O. 2013. Tide-surge-wave modelling and forecasting in the Mediterranean Sea with focus on the Italian coast. Ocean Model. 61:38–48.

Ferrarin C, Bellafiore D, Sannino G, Bajo M, Umgiesser G. 2018. Tidal dynamics in the inter-connected Mediterranean, Marmara, Black and Azov seas. Prog Oceanogr. 161:102–115.

Ferrarin C., Davolio S., Bellafiore D., Ghezzo M., Maicu F., Mc Kiver W., Drofa O, Umgiesser G., Bajo M., De Pascalis F., Malguzzi P., Zaggia L., Lorenzetti G. & Manfè G. 2019. Crossscale operational oceanography in the Adriatic Sea, J Oper Oceanogr., DOI:10.1080/1755876X.2019.1576275

Kourafalou V, Mey PD, Staneva J, Ayoub N, Barth A, Chao Y, Cirano M, Fiechter J, Herzfeld M, Kurapov A, et al. 2015. Coastal ocean forecasting: science foundation and user benefits. J Oper Oceanogr. 8(sup1):s147–s167.

Stanev EV, Grashorn S, Zhang YJ. 2017. Cascading ocean basins: numerical simulations of the circulation and interbasin exchange in the Azov - Black - Marmara - Mediterranean Seas system. Ocean Dyn. 67:1003–1025.

Wilkin J, Rosenfeld L, Allen A, Baltes R, Baptista A, He R, Hogan P, Kurapov A, Mehra A, Quintrell J, et al. 2017. Advancing coastal ocean modelling, analysis, and prediction for the US Integrated Ocean Observing System. J Oper Oceanogr. 10(2):115–126.

Zhang YJ, Stanev E, Grashorn S. 2016. Unstructured-grid model for the North Sea and Baltic Sea: validation against observations. Ocean Model. 97:91–108.

---

## Author Comment (AC1) · 26 Apr 2019

Many thanks to Reviewer-1 for his thorough revision and the number of useful tips. Please find attached a PDF document where a detailed point-by-point response is provided.

Please also note the supplement to this comment:
https://www.ocean-sci-discuss.net/os-2018-168/os-2018-168-AC1-supplement.pdf

---

## Author Comment (AC2) · 30 Apr 2019

In this manuscript, titled "Skill assessment of global, regional and coastal circulation forecast models: evaluating the benefits of dynamical downscaling in IBI surface waters", the authors describe an inter-comparison of different oceanographic operational systems in the Iberian shelf and in the Gibraltar Strait. Although the numerical results are deeply investigated showing the different model performance in reproducing Surface water properties, to my opinion the manuscript suffers from a lack of identification of the novel aspects of the research. Al stated by the authors, the three considered operational systems have been already described in many publications. Similarly, the benefits of downscaling (i.e. increasing resolution) in coastal waters have been widely treated in literature.

**Many thanks to Dr. Ferrarin for his detailed review and the number of useful tips that will aid to improve the paper. Please find below a detailed point-by-point response where each caveat has been carefully addressed with the hope of improving the quality of the document and make it acceptable for publication.**

**With regards to reviewer´s claim about the lack of identification of the novel aspects of the research, we would like to say that this paper serves two purposes: the first one is not new at all (*performing a multi-parameter model intercomparison from global to local scales to infer the added value of downscaling approaches*) but the second goal (*event-oriented model skill assessment and evaluation of models ability to accurately reproduce an extreme coastal process such as the surface flow reversals in the Strait of Gibraltar*) is, at least within Copernicus Marine Service (CMEMS) perspective, a rather novel theme. In particular, evaluating the ability of each model to reliability portray the AJ variability and the aforementioned episodic full reversals of the surface flow is an additional asset for wise decision-making of Algeciras Bay Harbor operators. A detailed characterization of this unusual phenomenon in the Strait of Gibraltar is relevant from diverse aspects, encompassing search and rescue operations (to adequately expand westwards the search area), the management of accidental marine pollution episodes (to establish alternative contingency plans), or safe ship routing (to maximize fuel efficiency). In this context, a new paragraph has been added in the introduction in order to highlight the novel aspects of this work.**

**Major comments:**

**1.** I appreciate the authors for highlighting the need of accurately addresses land-sea, air-sea, and coastal-offshore interactions when dealing with regional operational ocean systems (see also Kourafalou et al. 2015; Wilkin et al. 2017; Ferrarin et al., 2019). However, I think the authors should mention in the introduction that the downscaling from open sea (ocean) to coastal area can be also achieved through the implementation of numerical models based on a unique unstructured grid (Cucco et al. 2012; Ferrarin et al. 2013, 2019; Zhang et al. 2016; Federico et al. 2017; Stanev et al. 2017) able to describe processes at different spatial scales, and not only or through nesting of models. I'd like to point out that unstructured approach is particularly useful

for realizing a seamless transition between adjoining regional seas connected by narrow straits (i.e. the Gibraltar Strait; Stanev et al. 2017 and Ferrarin et al., 2018).

**Since we fully agree with this comment, a paragraph has been added to the introduction with the aim of clarifying that unstructured grid approaches constitute an useful and flexible alternative to complicated and computational time-demanding multiple nesting procedures. All the suggested works have been cited and subsequently added to the reference list.**

**2.** The only novel aspect in this work seems to be the inclusion of the so called "spectral" nudging technique into the operational IBI system. A similar methodology has been recently applied by Ferrarin et al. (2019) in the Tiresias operational system with a relaxation coefficient spatially varying over the Adriatic Sea domain (as a function of the grid resolution) from 2 days in the open sea and increasing, thus diminishing the restoration contribution, toward the coast. The authors should provide more details about the nudging methodology in terms of variables assimilated and 3D spatial weight functions (relaxation time).

**Further details about this methodology have been provided in section 3.2 in order to clarify the impact of spectral nudging on IBI forecast estimations.**

**The nudging is applied on a weekly basis during the analysis cycle but the increment applied (described below) is the weekly mean of the daily increments computed between the parent model analysis and the child model forecast.**

**After each forecast cycle of the child system (IBI), a corresponding analysis cycle is re-launched (Figure 1). The increment used to nudge IBI during the analysis cycle consists in a space and time low pass filter of the difference between the state variables XP (typically currents, temperature and salinity) of the parent model analysis and the variables XC of the child model forecast. As we want to keep the characteristic scales the parent GLOBAL system can resolve (mesoscale structures and the representation of these structures in terms of geostrophic turbulence are improved by the data assimilation of the sea level anomalies), the nested IBI system must be nudged by the parent resolved scales from large scale to mesoscale. So the space and time filter has to keep these scales and remove the smaller ones. Since a week is the typical characteristic time scale of the mesoscale structures, the chosen time smoothing consists in a simple weekly mean of the difference XP – XC, which is calculated in the parent system grid.**

[Figure]

*Figure 1: Spectral nudging. Delta is the increment of the i-cycle. IAU mean Incremental Analysis Update*

**The whole increment is applied during the analysis cycle at each time step, following the function g(t) shown in Figure 2. In order to prevent a discontinuity between the increments of two successive analysis cycles i-1 and i and also so to ensure the transition between two cycles, a 1-day overlap is applied which corresponds to the linear decrease in the weight on the increment for cycle i-1 and the linear increase in the weight on the increment for cycle i.**

[Figure]

*Figure 2: Function g(t)*

**Finally, here we present the nudging parameters applied (Table 1):**

| Simulation ID | Spectral nudging | Time window | Nudging Applying | | Spatial window |
|---|---|---|---|---|---|
| 1- Nudged run IBI-NUDG | Yes | 7 days | 1-
2-
3- | Bottom > 200 m
30km off coast
Depth > 3000m | 45 km |
| 2- Free run IBI-REF | No | ---- | | ---- | ----- |

*Table 1: Nudging parameters applied to the 'nudged IBI simulation' (with applying of spectral nudging), in contrast with the 'free IBI simulation' (i.e. without applying of spectral nudging).*

Within this context, we must emphasize that a tapering function is used in IBI operational chain: this space weight function can be compared to a 3D mask with transition between the zones of the nested system which have to be nudged - typically in open waters, outside the continental shelf area- and which remain free - continental shelf, coastal areas and regions close to the open boundaries- (Figure 3). In those regions where the tapering function is zero (blue color in Figure 3), no spectral nudging is applied and IBI system runs freely. By contrast, in those regions where the tapering function is set to one (red color in Figure 3), the spectral nudging is applied and therefore IBI performance is rather similar to its parent system, the GLOBAL. Both model performances are not absolutely identical in open-waters because the aforementioned weekly increment is a weighted average where a time and space low pass filter is applied to the differences between the state variables XP and XC in order to keep the scales better resolved by GLOBAL and discard the smaller ones.

[Figure]

*Figure 3: Tapering function for IBI forecasting system. Left panel: a section at 46°N of the total tapering function. Right: tapering function at surface.*

For further details, we refer the reviewer to Herbert et al. (2014).

Herbert, G., Garreau, P., Garnier, V., Dumas, F., Cailleau, S., Chanut, J., Levier, B., and Aznar, R.: Downscaling from oceanic global circulation model towards regional and coastal model using spectral nudging techniques: application to the Mediterranean Sea and IBI area models, Mercator Ocean Quarterly Newsletters, 49, 44, April 2014.

Moreover, it is not clear what are the benefit of this improvement, since according to the results presented in Fig 2-5, IBI performance is lower to the GLOBAL model on open sea regions (IBISR, ICANA, WSMED), where ideally nudging allows the model state to be reconciled with the assimilated GLOBAL data.

We have thoroughly revised Figure 2 of the manuscript, which was directly extracted from NARVAL automatic validation web tool. We have detected that IBI panels for May 2017 (Figure 2-f) and July 2017 (Figure 2-h) are wrong. Since there was an issue related to bad atmospheric forcings from April to July 2017 (the solar radiation flux presented anomalously high values), IBI free simulations during this period were seriously affected and thus the computation of increments (described here in Figure 1) to later perform the spectral nudging was inaccurate, giving rite to the SST overestimation observed in those two panels of Figure 2 of the manuscript (f and h).

Once the issue was detected in July 2017 and the atmospheric forcing was fixed, IBI simulations were re-launched and new consistent outputs were uploaded to the CMEMS online catalog, replacing the wrong files. End-users were timely informed about this change in the catalog though the CMEMS service desk (for further details, see attached report entitled "CMEMS_RFC" at the end of the present document).

However, as NARVAL validation tool works automatically, monthly figures and skill metrics were routinely computed as soon as the wrong simulations ended. We should have launched manually the NARVAL toolbox after finishing the right simulations in July 2017 for those four previous months impacted by the corrupted atmospheric forcing in order to properly update NARVAL tool with the right monthly panels and metrics for the period April-July 2017, but we did not.

We apologize for any inconvenience derived from this lapse. Once NARVAL has been correctly updated, we have accordingly recomputed Figures 2 and 3 of the manuscript. Served as example, Figure 4 below exhibits the main changes introduced, where GLOBAL and IBI performances look rather alike in open waters (Figure 4-a and 4-c, respectively), highlighting the benefits of the spectral nudging technique. As previously indicated, both model performances are very similar but not absolutely identical in open-waters because the weekly increment is a weighted average where a time and space low pass filter is applied to the differences between the state variables XP and XC.

[Figure]

Figure 4: Monthly SST bias (model minus observation) directly extracted from NARVAL validation web tool. a) GLOBAL versus L3; b) IBI versus L3 (wrong dataset, generated with bad atmospheric forcings); c) corrected maps of IBI versus L3, after fixing atmospheric forcings: overall improvement in IBI performance over the entire domain.

**In this context, Figures 2 and 3 of the manuscript have been replaced by a new ones in order to better demonstrate how IBI benefits from the data assimilation into GLOBAL by means of the spectral nudging technique. Instead of randomly selecting specific months of 2017 (old Figure 2), the new Figure 2 of the manuscript is focused on the annual (2017) SST bias of GLOBAL and IBI predictions against L3 satellite observations:**

[Figure]

Figure 5: a) Availability of L3 data for 2017. Annual comparison of SST between GLOBAL (b) and IBI (c) against satellite-derived L3 data.

Complementarily, in order to provide further insight into the results derived from the spectral nudging technique, here we show the annual (2017) mean absolute difference (MAD) obtained for GLOBAL and IBI in open waters (where the spectral nudging is activated: Figure 6, a-b) and also for coastal waters (where there is no spectral nudging: Figure 6, c-d).

[Figure]

*Figure 6: Annual (2017) Mean Absolute Difference (MAD) of SST derived from the comparison of GLOBAL (left) and IBI (right) against L3 satellite estimations. a-b) Open waters; c-d) Coastal waters. Skill metrics gathered in the right black box.*

According to the skill metrics (right box in Figure 6), spatially-averaged over the entire open waters domain, GLOBAL performance is slightly better in open waters due to the direct data assimilation scheme. IBI benefits indirectly from the data assimilation implemented in the parent system through the spectral nudging (MAD = 0.15°) and even outperforms GLOBAL locally in specific zones, delimited with blue rectangles in Figure 7-a, such as the continental shelf break, western Canary Islands or a portion of the African upwelling system. By contrast, GLOBAL outperforms IBI in the Gulf of Cadiz and the NW Iberian open waters (Figure 7-b).

Equally, based on the skill metrics spatially-averaged over the entire coastal waters domain, IBI performance is, on average, better than GLOBAL (0.17° versus 0.20°) in coastal areas thanks to several factors (among others, the higher horizontal

resolution). IBI clearly outperforms GLOBAL in the continental shelf (English Channel, Irish and North Sea) and in the Strait of Gibraltar (Figure 7-c), although it is also true that GLOBAL is slightly more accurate in some parts of the Iberian and African upwelling systems (Figure 7-d).

[Figure]

*Figure 7: Idem to Figure 8, but zones of worse model performance (i.e., higher MAD) are delimited with blue rectangles.*

The MAD metrics for each subregion within IBI domain (defined in Figure 1-a of the first version of the manuscript) are gathered below in a table and confirm what we above speculated: overall, IBI outperforms GLOBAL is those subregions where no spectral nudging is applied (English Channel, Irish and North Sea, North Iberian shelf) except in the Western Iberian Shelf (WIBSH). By contrast, better metrics are obtained for IBI in the Gulf of Biscay (GOBIS), where the spectral nudging is only applied in the westernmost zone.

On the contrary, GLOBAL performance is generally more accurate in open waters thanks to the direct data assimilation. IBI metrics, although are consistent, are slightly worse than those obtained for GLOBAL.

| Region | Spectral Nudging applied? | MAD GLO-L3 (°) | MAD IBI-L3 (°) |
|---|---|---|---|
| IRISH | NO | 0.22 | 0.16 |
| ECHAN | NO | 0.24 | 0.19 |
| GOBIS | Partially applied in some zones | 0.15 | 0.14 |
| NIBSH | NO | 0.21 | 0.17 |
| WIBSH | NO | 0.20 | 0.24 |
| CADIZ | Partially applied in some zones | 0.23 | 0.27 |
| WSMED | Partially applied in some zones | 0.17 | 0.20 |
| GIBST | NO | 0.34 | 0.28 |
| ICANA | Partially applied in some zones | 0.19 | 0.23 |

*Table 2: Annual (2017) Mean Absolute differences (MAD) of Sea Surface Temperature (SST) of GLOBAL and IBI against L3 satellite data for nine different subregions defined within IBI regional domain. Metrics highlighted in green (red) denote a better (worse) model performance.*

Do the differences with the GLOBAL demonstrate that over the sea surface the impact of the air-sea heat fluxes is stronger that the restoration contribution of the GLOBAL nudging?

Once we have fixed the issue previously commented (corrupted atmospheric files with anomalously high solar radiation flux during April-July 2017), the differences between GLOBAL and IBI performances are reduced (especially in open-waters) as previously exposed in this document. Served as additional example, below we present the seasonal maps of bias for the SST. These results highlight that air-sea heat fluxes, albeit relevant, play a secondary role respect to the primary influence of the restoration contribution of the GLOBAL nudging.

[Figure]

*Figure 8: Seasonal comparison of SST between GLOBAL (b) and IBI (c) against satellite-derived L3 data.*

In this context, the authors should demonstrate the benefit of the nudging approach analyzing the water column properties and not only the sea surface.

**Since the present work is entitled "*Skill assessment of global, regional and coastal circulation forecast models: evaluating the benefits of dynamical downscaling in IBI surface waters*", the reviewer´s suggestion about analyzing the water column properties, albeit valuable, is perhaps beyond of the scope of this contribution and deserves further attention in the frame of other future paper. However, we agree with the reviewer that an additional insight into the benefits of the spectral nudging technique on the surface must be provided in the second version of the manuscript we are currently preparing. According to the new results obtained and above exposed, we consider that the added-value of the spectral nudging on the surface has been proved.**

**3.** One of the most important topic described in the manuscript is the model comparison in the Gibraltar Strait. As one would expect, the SAMPA high resolution model performs better in reproducing hydrodynamics in this region having complex morphology and dynamics. However, it is not clear to me if this improvement is only due to the model spatial resolution or to the inclusion of the remote effect of the atmospheric forcing in the barotropic flow thought the strait.

**We fully agree that further stress should be made in the discussion section about the role of barotropic flows. To this aim, the following paragraph has been added to section 6:**

**"The reversal of the surface inflow is caused by meteorological-driven flows through the Strait associated with the passage of high pressure areas over the Mediterranean (García-Lafuente et al. 2002). Because these flows originate in the far field and not in the Strait itself, the different grid resolution of IBI and SAMPA do not appear the likely explanation for these events to do not show up in the IBI model. Instead, their different skill in capturing such extreme events seems to be associated to their different forcing."**

**In consequence, several parts of the manuscript have been modified and enhanced in order to better clarify the relevance of the atmospheric (remote) forcing in the flow reversals and how sub-tidal barotropic lateral forcing for SAMPA, obtained from the NIVMAR storm surge model, ensures that the SAMPA model captures a realistic variability of inflow and outflow currents though the Strait.**

According to the model description reported at page 7, SAMPA is not directly nested in IBI since it is forced by daily fields (from IBI) and tidal and meteorological-driven barotropic velocities from MOG2D and NIVAR. In this context the authors should:

**3.1.** make clear throughout the whole manuscript that SAMPA is not only nested in IBI. Therefore, IBI could not be defined the only parent system of SAMPA.

**In the new version of the manuscript, we have clearly stated that SAMPA is not only nested to IBI but also forced by tidal and meteorological-driven barotropic velocities from MOG2D and NIVAR, respectively. This fact has been reflected not only in Table 1 but also in several sections of the document.**

**3.2.** Explain the reason of this choice, which sound counter-intuitive to me, since IBI already account for tide and meteorological forcing.

**SAMPA does not incorporate tides from IBI (i.e., storm surge) because by the time SAMPA was implemented, IBI products were only available on daily averages. Such storage frequency was not suitable to resolve meteorologically-driven (barotropic) flows through the Strait either, as the associated currents vary at a typical time scale of 3-4 days (García-Lafuente et al. 2002). Hence, tidal and sub-tidal barotropic lateral forcing for SAMPA are respectively obtained from published tidal constituents (Carrere and Lyard 2003) and from the NIVMAR storm surge model (Álvarez-Fanjul et al. 2001). Because NIVMAR has been validated against mooring observations in the Strait of Gibraltar (Álvarez-Fanjul et al. 2001), this nesting strategy ensures that the SAMPA model captures a realistic variability of inflow and outflow currents. As part of possible next improvements of the SAMPA system, we will explore the possibility of a more direct nesting strategy into IBI since the current operational version of IBI includes the delivery of 3D hourly outputs.**

**In order to clarify this question, several sentences have been introduced in section 3.3 devoted to describe SAMPA system and also in the conclusions.**

**References:**

**García Lafuente, J., Álvarez Fanjul, E., Vargas, J. M., and Ratsimandresy, A. W., Subinertial variability in the flow through the Strait of Gibraltar, J. Geophys. Res., 107( C10), 3168, doi:10.1029/2001JC001104,2002."**

**Álvarez-Fanjul., E., Pérez, B. and Rodríguez, I.: Nivmar: a storm surge forecasting system for Spanish waters, Sci. Mar., 65, 145-154, 2001.**

**3.3.** the authors should discuss the results of the inter-model comparison in the light of the different approaches adopted for the forcing and parametrization in the three systems, not only resolution.

**The authors absolutely agree with this comment. Although it was addressed in the conclusions, it is also true that we should have emphasized more times along the entire document (especially in the discussion of results) that increased resolution is not the only factor to explain a more accurate model performance.**

**As the reviewer will see, several parts of the manuscript have been modified and enhanced in order to better clarify the relevance of the atmospheric (remote) forcing in the flow reversals and how sub-tidal barotropic lateral forcing for SAMPA, obtained from the NIVMAR storm surge model, ensures that the SAMPA model captures a realistic variability of inflow and outflow currents though the Strait.**

**Therefore, the remote barotropic effect of the meteorological forcing over the Mediterranean Sea, which is not included in IBI and GLOBAL, play a major role in the regulation of the seasonal cycle of the AJ and of its occasional inversions.**

**Minor comments:**

**Page 2, line 35**: Is Herbert et al., 2014 (on Mercator Ocean - Quarterly Newsletter) a peer review publications? Only peer review articles should be cited.

**No, it is not. We fully agree that only peer review articles should be cited but, in this specific case, we would like to make one single exception since Herbert et al. (2014) provides further insight into the spectral nudging strategy here adopted and could be of special concern for some readers. To author´s knowledge, this is the only existing work published where the spectral nudging adopted in IBI system is extensively addressed. For this reason, we would like to keep it in the reference list (if the reviewer allows us).**

**Reference:**

**Herbert, G., Garreau, P., Garnier, V., Dumas, F., Cailleau, S., Chanut, J., Levier, B., and Aznar, R.: Downscaling from oceanic global circulation model towards regional and coastal model using spectral nudging techniques: application to the Mediterranean Sea and IBI area models, Mercator Ocean Quarterly Newsletters, 49, 44, April 2014.**

**Page 3, line 20**: Sotillo et al., 2018. Do not include under review article.

**Done!**

**Page 3, lines 22-29**: The part regarding operational products for harbours seams out of topic.

**The operational implementation of SAMPA coastal model was led by Algeciras Bay Port Authority and Puertos del Estado in order to provide useful tools for wise decision-making and efficient coastal management in a region with intense maritime traffic and the often harsh met-ocean conditions. In order to reflect this fact, a new sentence has been added to section 3.3 where SAMPA system is described.**

**"*The PdE-SAMPA model application was developed by the University of Malaga in collaboration with PdE in order to provide a tailored forecasting service to one of their main stakeholders, the harbor of Algeciras Bay (Figure 1-b).*"**

**Moreover, when adopting dynamical downscaling from global/regional scales to coastal scales, port approach areas constitute the last step in this phased step-wise approach since tailored forecasting products with extremely fine grid resolution (up to 50 m) are required for safe and sound harbor operations in support of blue growth. Therefore, we humbly consider that the aforementioned paragraph is aligned with the topic addressed in this manuscript.**

**Page 5, line 8**: Please provide a reference for this statement.

**Done! The paragraph has been modified according to the reviewer´s suggestion:**

**The Strait of Gibraltar (GIBST), the only connection between the semi-enclosed Mediterranean basin and the open Atlantic Ocean (Figure 1, b), is characterized by a two-layer baroclinic exchange which is hydraulically controlled at Camarinal Sill (Sánchez-Garrido et al., 2011).**

**Where:**

**Sánchez-Garrido, J.C., Sannino, G., Liberti, L., García Lafuente, J. and Pratt, L.: Numerical Modeling of three-dimensional stratified tidal flow over Camarinal Sill, Strait of Gibraltar. Journal of Geophysical Research, 116, C12026, doi: 10.1029/2011JC007093, 2011.**

**Page 6, lines 10-24**: Provide more details about the nudging approach.

**As previously indicated in point 2 of section "Major comments", further details about the spectral nudging approach adopted in the operational chain have been provided in section 3.2 of the manuscript.**

**Page 8, lines 13-14 and 38-41**: see major comment 3.

**As previously indicated in point 3 of section "Major comments", we have added some pieces of text throughout the whole manuscript to unequivocally state that SAMPA coastal model is not only nested in IBI but also to NIVMAR and Mog2D.**

**Page 10**: Section Temperature should be shortened.

**The authors are fully aware that this sub-section is too extensive (almost 3 pages). In order to solve this issue and also to provide further details about the benefits of the spectral nudging (that were not adequately exposed in the first version of the manuscript), section temperature has been refocused and shortened.**

**Page 12, line 6**: "GLOBAL performance was more accurate from July April to December as indicated by the lower RMSD".

**As previously indicated, Figures 2 and 3 of the first version of the manuscript were impacted by the reported issue, related to bad atmospheric forcings from April to July 2017. Once the atmospheric forcing was fixed, IBI simulations were re-launched and new consistent outputs were obtained. Accordingly, Figures 2 and 3 have been replaced and the text has been updated.**

**Page 12, line 28-29**: This is not true, since divergence between IBI and GLOBAL forecast are found also in the open sea.

**We have replaced the sentence "divergencies are mainly found on the shelf near coastal areas" by other one to highlight that differences were also detected in open-waters.**

**Page 16, line 44**: ": : : midpoint of the selected transect (white square in Fig. 1c)."

**True! The reference to Figure 1-c has been added to illustrate where the midpoint is located.**

**Reference:** authours should limit self referencing (29/76) and include only peer reviewed articles without considering under review works (Sotillo el al., 2018), conference abstract (Hernández-Lasheras et al., 2018) or poster (Lorente et al., 2017).

**Done! Only peer-reviewed articles have been cited. Other types of contributions have been removed from the reference list, except Herbert et al. (2014) for the reasons above exposed.**

**Page 24, line 19**: provide doi for "in press" article.

**Done! Doi, volume and pages have been provided for this article.**

**Page 24, line 32:** provide the full list of authors.

**Done! The entire list of authors has been provided.**

**Page 26, line 28**: the journal for this reference is missing.

**Thanks for finding this typo. Part of the title was missing. Now the reference is correct.**

**Figure 1:** please indicate the SAMPA model domain; specify the meaning of the black line and white square in panel C.

**Done! Figure 1-b already denotes the coverage domain of SAMPA model, although we forgot to explicitly mentioning it in the Figure caption. We have modified both the figure and the caption to clearly illustrate the SAMPA domain.**

**Table 2:** Open boundary conditions for SAMPA should also include MOG2D.

**True! The section "Open boundary condition" from Table 1 (not Table 2) has been updated with this relevant information. It is also true that such details were already provided below in the same Table 1 in the section "Tidal forcing", so perhaps there is some redundancy. Finally, in section 3.3 (SAMPA model description) we have clarified:**

**[...] tidal and meteorologically-driven barotropic velocities are prescribed across the open boundaries, the former extracted from the Mog2d model described by Carrere and Lyard (2003) and the latter from the storm surge operational system developed by Álvarez-Fanjul et al. [...]**

**Suggested references:**

Cucco, A., Sinerchia, M., Ribotti, A., Olita, A., Fazioli, L., Perilli, A., Sorgente, B., Borghini, M., Schroeder, K. and Sorgente, R.: A high-resolution real-time forecasting system for predicting the fate of oil spills in the Strait of Bonifacio (western Mediterranean Sea). Marine Pollution Bulletin, 64 (6), 1186–1200, 2012.

Federico, I., Pinardi, N., Coppini, G., Oddo, P., Lecci, R. and Mossa, M.: Coastal ocean forecasting with an unstructured grid model in the southern Adriatic and northern Ionian seas. Nat. Hazards Earth. Syst. Sci., 17(1), 45–59, 2017.

Ferrarin, C., Roland, A., Bajo, M., Umgiesser, G., Cucco, A., Davolio, S., Buzzi, A., Malguzzi, P. and Drofa, O.: Tide-surge-wave modelling and forecasting in the Mediterranean Sea with focus on the Italian coast. Ocean Modelling, 61, 38–48, 2013.

Ferrarin, C., Bellafiore, D., Sannino, G., Bajo, M. and Umgiesser, G.: Tidal dynamics in the inter-connected Mediterranean, Marmara, Black and Azov seas. Progress in Oceanography, 161, 102–115, 2018.

Ferrarin, C., Davolio, S., Bellafiore, D., Ghezzo, M., Maicu, F., Mc Kiver, W., Drofa, O., Umgiesser, G., Bajo, M., De Pascalis, F., Malguzzi, P., Zaggia, L., Lorenzetti, G. and Manfè, G.: Cross-scale operational oceanography in the Adriatic Sea. Journal of Operational Oceanography, doi:10.1080/1755876X.2019.1576275, 2019.

Kourafalou, V., Mey, P.D., Staneva, J., Ayoub, N., Barth, A., Chao, Y., Cirano, M., Fiechter, J., Herzfeld, M., Kurapov, A., Moore, A.M., Oddo, P., Pullen, J., Van der Westhuysen, A. and Weisberg, R.H.: Coastal ocean forecasting: science foundation and user benefits. Journal of Operational Oceanography, 8 (S1), s147–s167, 2015.

Stanev, E.V., Grashorn, S. and Zhang, Y.J.: Cascading ocean basins: numerical simulations of the circulation and interbasin exchange in the Azov - Black - Marmara - Mediterranean Seas system. Ocean Dynamics, 67, 1003–1025, 2017.

Wilkin, J., Rosenfeld, L., Allen, A., Baltes, R., Baptista, A., He, R., Hogan, P., Kurapov, A., Mehra, A., Quintrell, J., Schwab, D., Signell, R. and Smith, J.: Advancing coastal ocean modelling, analysis, and prediction for the US Integrated Ocean Observing System. Journal of Operational Oceanography, 10 (2), 115–126, 2017.

Zhang, Y.J., Stanev, E.V. and Grashorn, S.: Unstructured-grid model for the North Sea and Baltic Sea: validation against observations. Ocean Modelling, 97, 91–108, 2016.

**All the suggested references have been incorporated to the list (except Federico et al. 2017 and Kourafalou et al., 2015 since they were both already included in the first version of the manuscript) and accordingly cited in the document. Many thanks to the reviewer for these useful recommendations to enrich the manuscript.**

**Copernicus Marine Environment Monitoring Service Request For Change**

| | |
|---|---|
| **Change**            **Notification** | **Your Reference Number** MYO-1137 |
| **Risk Level**   1 | **Change Type**   Standard |
| **Summary of the change**
Update of the IBI-MFC PHY & BIO historical timeseries in order to substitute files affected by a temporal bug occurred during the first fortnight of April in the atmospheric forcing files. This opportunity will be also used to include two attributes in the header of the files of the BIO product that are necessary for their right display with the GODIVA viewer.

Affected products:
IBI_ANALYSIS_FORECAST_PHYS_005_005
IBI_ANALYSIS_FORECAST_BIO_005_004 | **Submission Date**      20170725 |
| | **Requester (Name)**         Arancha Amo Baladrón |
| | **Requester (CMS Element)**       IBI-MFC/Puertos del Estado PC |
| | **Requester (Email)**        aab@puertos.es |

**Justification for change** (continue on additional sheet or attach document as necessary)

TASK 1:
Due to a temporal bug that affected the preprocess of the atmospheric forcing files the first fortnight of April, the solar radiation flux presented slightly higher values than expected during that period. The IBI-MFC PHY and BIO hindcast runs (the ones executed once a week to generate the historical timeseries) were forced with this solar radiation flux data. Therefore, in order to supply the best possible solution to the IBI-MFC users, the affected historical timeseries have been generated again from 5th of April 2017 using the right solar radiation flux forcing files. Products delivered from August 2nd onwards will not be affected by this issue.

TASK 2:
As the whole BIO historical timeserie needs to be updated for this task, a minor bug that only affects to the display through the GODIVA viewer of two variables of the IBI_ANALYSIS_FORECAST_BIO_005_004 product (the silicate and the dissolved molecular oxygen concentrations) will be fixed.
This display tool requires the presence of the "scale_factor" and "offset" attributes in the netcdf header for every variable. This information is usually stored in the headers of the netcdf files in the IB-MFC operational suites, however, when these attributes take values 1 and 0 for "scale_factor" and "offset", respectively, they are not saved. This was the case for the aforementioned variables. In order to fix this minor bug, only an update of the code to force the creation of the missing attributes for both variables in the header of the netcdf files is needed.

| **Quality Likely Impact** (continue on additional sheet or attach document as necessary – Reminder: if product quality impacted, the related QuID should be updated and delivered)

TASK 1: Increase in the quality of the IBI products, as the update involves the use of improved forcing files.
TASK 2: No change in the quality of the IBI products, as this task only affects to the header information. | **Technical Likely Impact** (continue on additional sheet or attach document as necessary)
**Warning : Please ensure the DU for the related product is informed of the change**
TASK 1: No change in the IBI product data content, only in its quality.
TASK 2: No change in the IBI product data content, only in the header of two BIO variables.
No other CMEMS systems/serviced will be affected.
A maximum outage of 2 hours in the access to the user interfaces might be produced. |

CMEMS Request For Change – V1.0 June 2015

| | Affected product:
IBI_ANALYSIS_FORECAST_PHYS_005_005
IBI_ANALYSIS_FORECAST_BIO_005_004 |
|---|---|

**Description of risks and mitigation activities** (continue on additional sheet or attach document as necessary)
There are no big risks associated to this update.
However, in case of detecting any failure due to the modifications introduced in the code or in the update of the historical data, a backup of the old operational suite and the netcdf files of the historical timeseries is available and can be used to recover them.

| Implementation plan
(continue on additional sheet or attach document as necessary)
The implementation plan of this update is coordinated with the IBI IT-Team.
Both historical timeseries, PHY & BIO, have been already re-generated.
The date proposed for the update of the PHY & BIO data is the Wednesday 2nd of August between 11:00-13:00UTC.
A PSO will be sent to the CMEMS Service Desk for the possible outages during the implementation of this RFC. | Backout plan
(continue on additional sheet or attach document as necessary)
In case of detecting any failure due to the modifications introduced in the code or during the update of the historical data, a backup of the old operational suite and the netcdf files of the historical timeseries is available and can be used to recover them. |
|---|---|
| Test plan
(continue on additional sheet or attach document as necessary)
The updated data have been validated against observational sources resulting in an improvement of the quality for these products. On the other hand, the right display of all the variables of the BIO product affected by the missing attributes has been checked using the GODIVA viewer. | Communication Plan
(continue on additional sheet or attach document as necessary)
CMEMS Service Desk is informed about this RFC. |

| **Related tickets** | **Scheduled Start Date / Time (UTC)**
2017-08-02  / 11:00 | **Scheduled Finish Date / Time (UTC)**
2017-08-02 / 13:00 |
|---|---|---|
| **Decision**: Accepted/Refused | **Decision on implementation schedule**: | |

---

## Author Response (AR2)

**Response to referee Dr. Ferrarin**

The authors improved significantly this revised version of the manuscript, addressing adequately and carefully the reviewers concerns. I particularly enjoyed reading the paper, which is clear, to the point and most interesting.

I've only some minor comments:

- In the Abstract (and also in other sections, eq. Pag. 17) the authors states that "SAMPA appeared to better reproduce the reversal events detected with HFR estimations, demonstrating the added value of imposing accurate meteorologically-driven barotropic velocities in the open boundaries (imported from NIVMAR storm surge model) to take into account the remote effect of the atmospheric forcing over the entire Mediterranean basin, which was not included in IBI and GLOBAL systems". I think the last part of the sentence is not entirely correct since the GLOBAL model represents the Mediterranean Sea and at least considers the barotropic effect of the wind forcing (not clear if it also considers the air pressure effect). IBI also considers the daily barotropic atmospheric forcing driven by the wind, being nested in GLOBAL, and partially the inverse barometric effect over the (Western) Mediterranean. Please consider reformulating the mentioned statements. Since we fully agree with this comment, we have accordingly modified both the abstract and the main body of the manuscript (page 17).

Abstract: "which was only partially included in IBI and GLOBAL systems"

Page 17, a new paragraph has been added:

"Since the atmospheric pressure forcing is missing in the GLOBAL sea level outputs, this system only considers the barotropic response to wind forcing. In the case of IBI system, both atmospheric forcings are taken into account but the inverted barometer approximation is solely imposed over the Western Mediterranean (not over the entire basin). Therefore, only a portion of the subtidal variability of the flow through the Strait of Gibraltar can be adequately explained (García-Lafuente et al., 2002)."

- Page 7: I suggest to specifically mention that the lateral boundaries of the IBI model are both the Atlantic Ocean and the Mediterranean Sea.

**Done!**

- Page 8, line 27: In the two lateral open boundaries (west - the Atlantic Ocean, east - the Western Mediterranean Sea) ...

**Done!**

- Page 21. line 37: This coastal ...

**Done!**

- Page 24, line 9: I suggest to include the web address of the mentioned Inmerse H2020 project.

**Done!** In addition, the acronym IMMERSE has been described. - Figure 2: Remove "Spectral nudging on" and "Spectral nudging off" from panels b and d since the GLOBAL model does not have nudging. Done!

- Figure 12: For better readability, consider using lines instead of points in panel b.

Since the other referee (Dr. Macias) asked for the opposite modification (from lines to points) during the first revision, we consider that Figure 12 should be kept as it is right now.